# ProteinWeaver: A Divide-and-Assembly Approach for Protein Backbone Design

## Abstract

Nature creates diverse proteins through a 'divide and assembly' strategy. Inspired by this idea, we introduce ProteinWeaver, a two-stage framework for protein backbone design. Our method first generates individual protein domains and then employs an SE(3) diffusion model to flexibly assemble these domains. A key challenge lies in the assembling step, given the complex and rugged nature of the inter-domain interaction landscape. To address this challenge, we employ preference alignment to discern complex relationships between structure and interaction landscapes through comparative analysis of generated samples. Comprehensive experiments demonstrate that ProteinWeaver: (1) generates high-quality, novel protein backbones through versatile domain assembly; (2) outperforms RFdiffusion, the current state-of-the-art in backbone design, by 13% and 39% for long-chain proteins; (3) shows the potential for cooperative function design through illustrative case studies. To sum up, by introducing a 'divide-and-assembly' paradigm, ProteinWeaver advances protein engineering and opens new avenues for functional protein design.

## 1 Introduction

Nature employs a sophisticated 'divide and assemble' strategy to create large and intricate protein structures that meet diverse biological functional needs (Fig. 1A) (Pawson & Nash, 2003; Huddy et al., 2024; P Bagowski et al., 2010). This process primarily involves the recombination of existing structural blocks, particularly protein domains, which serve as the fundamental, recurring units in protein structures. Remarkably, a limited number of protein domains (approximately 500 as classified in CATH) suffice to create more than hundreds of thousands of structures satisfying a wide array of functions (Orengo et al., 1997). This strategy enables the creation of multi-domain protein backbones, facilitating the emergence of cooperative functions.

Recent advances in protein backbone design have enabled the generation of novel and diverse structures with high designability (Watson et al., 2023; Ingraham et al., 2023; Yim et al., 2023; 2024; Bose et al., 2023; Lin & AlQuraishi, 2023; Lee et al., 2022; Wu et al., 2024a). However, our analysis reveals a significant limitation: designability decreases markedly as the backbone length increases (Fig. 1E). This limitation may stems from inadequate inter-domain interaction, evidenced by a dramatic decrease in domain numbers and interface scTM compared to native proteins (Fig.9 and Fig.1F). These findings highlight the need for an approach to address the intricacies of multi-domain organization in backbone design, particularly for larger protein backbones.

In this study, inspired by nature's strategies, we present ProteinWeaver (Fig.1B). Our method addresses the limitations of current approaches by breaking down the complex problem of backbone design into manageable components, framing it as a divide and assembly problem. ProteinWeaver operates in two stages (Fig. 1B): (1) Domain generation: We first divide the long sequence into multiple domains, focusing on generating these local structures independently, allowing for stable and accurate generation of individual domains; (2) Flexibly assembly: we employ a SE(3) diffusion model to learn the flexible assembly between these domains (Fig. 1C). This stage aims to capture the crucial inter-domain interactions. In short, this two-stage approach represents a brand-new paradigm for de novo backbone design.

The second assembly stage of our approach presents a significant challenge: designing optimal inter-domain interactions through precise weaving of independently generated protein domains. This

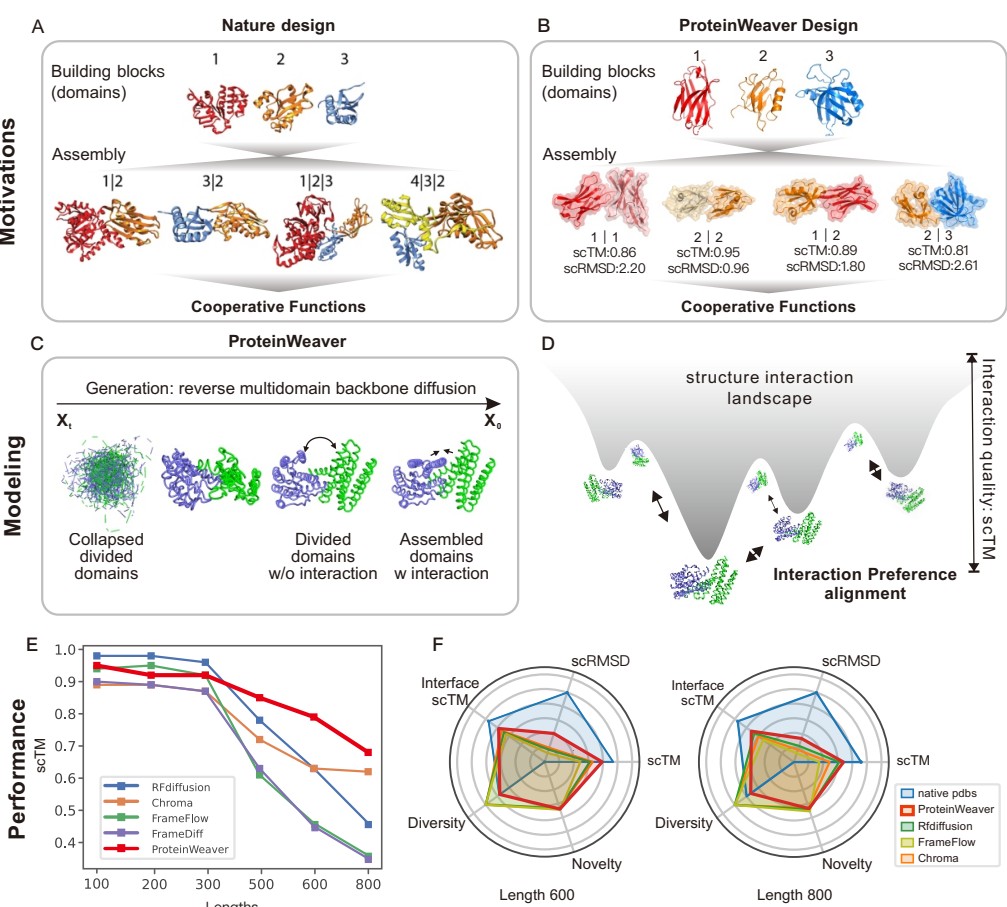

Figure 1: Overview of ProteinWeaver. **(A)** An illustration demonstrating the 'divide-and-assembly' approach to native protein evolution, which enhances cooperative function design. The pictures are adapted from this study (Aziz & Caetano-Anollés, 2021). **(B)** ProteinWeaver emulates natural strategies to create protein backbones. **(C)** ProteinWeaver is a backbone diffusion model. **(D)** The inter-domain structure-interaction landscape is complex and rugged, where minor structural modifications can lead to significant changes in interactions. Preference alignment technique aids in navigating this landscape effectively. **(E)** Existing methods struggle with long-chain backbone design, whereas ProteinWeaver demonstrates a considerable advantage. **(F)** A radar chart illustrates ProteinWeaver's overall performance in long-sequence backbone design. Inter-domain quality is evaluated using interface scTM metrics.

process requires the model to navigate a complex landscape of potential structural arrangements and their corresponding interactions at a fine-grained level (Kuhlman & Bradley, 2019; Maguire et al., 2021). Two primary factors pose difficulties: (1) The scarcity of domain interaction data, which limits the model's capacity to capture the fine-grained structure-interaction landscape. (2) The absence of an efficient method to explore the relationship between structure and interaction energy, as folding methods such as Alphafold2 are computationally intensive and time-consuming for model optimization purpose (Jumper et al., 2021).

To address these challenges, we define domain assembly as a structure-interaction landscape optimization problem and implement preference alignment. Our approach consists of two key steps. We first conducted extensive sampling to generate diverse multi-domain structures and quantitatively evaluate their interaction quality using scTM metrics. This process captures the complex distribution of the structure-interaction energy landscape in a fine-grained manner (Fig. 1D). Then, we implemented preference alignment. Rather than learning mappings to predefined labels in conven-

tional supervised fine-tuning (SFT), preference alignment enables our structure diffusion model to optimize through pairwise comparative analysis of the sampled structures, allowing the model to effectively navigate the intricate relationship between structure and interaction energy landscapes.

Our primary contributions can be summarized as follows:

(1) We propose the first 'divide-and-assembly' two-stage generation framework for protein backbone design. ProteinWeaver enables flexible assembly of general protein domains, allowing for the creation of sophisticated structures.

(2) We adopt the preference alignment technique to effectively navigate the domain interaction landscape and generate interaction-reasonable backbones in our diffusion model, which may provide a general approach benefiting backbone design.

(3) We present an extensive experimental evaluation of ProteinWeaver by comparing its performance with the state-of-the-art methods including RFdiffusion (Watson et al., 2023), Chroma (Ingraham et al., 2023), FrameDiff (Yim et al., 2023), and FrameFlow (Yim et al., 2024). ProteinWeaver significantly outperforms RFdiffusion by 13% to 39% in the quality of long-chain backbones (Fig. 1E) and exhibits comprehensive advantages across various metrics (Fig. 1F).

(4) We present ProteinWeaver's potential applications for cooperative function design through targeted domain assembly, as illustrated by case studies in substrate-directed enzyme design and bispecific antibody engineering.

## 2 PROTEINWEAVER

In this section, we introduce the ProteinWeaver. The overall generation framework is introduced in Sec.2.1. Details for training and sampling are introduced in Sec.2.2 and Sec.2.3.

### 2.1 DIVIDE AND ASSEMBLY DIFFUSION FRAMEWORK

**Protein backbone representation.** Following AlphaFold2 (Varadi et al., 2022), the structure of protein backbone is parameterized as a collection of rigid frames. For a protein backbone of length $L$, these frames are denoted by $\mathbf{T} = [T_1, T_2, ..., T_L] \in \mathrm{SE}(3)^L$. Each frame $T_i = (r_i, x_i)$ where $r_i \in \mathrm{SO}(3)$ and $x_i \in \mathbb{R}^3$ maps a rigid transformation from fixed, idealized coordinates $\mathrm{N}^*, \mathrm{C}_\alpha^*, \mathrm{C}^*, \mathrm{O}^* \in \mathbb{R}^3$, with $\mathrm{C}_\alpha^* = (0, 0, 0)$. Thus, for each residue $i \in [1, 2, ..., L]$, $\mathbf{S}_i = [\mathrm{N}, \mathrm{C}_\alpha, \mathrm{C}, \mathrm{O}]^i = T_i \circ [\mathrm{N}^*, \mathrm{C}_\alpha^*, \mathrm{C}^*, \mathrm{O}^*] \in \mathbb{R}^{4 \times 3}$. To construct the backbone oxygen O, we use an additional torsion angle $\psi$ by rotating $\mathrm{O}^*$ around the bond between $\mathrm{C}_\alpha$ and C. Finally, the complete 3D structure coordinates with all heavy atoms of a protein is denoted as $\mathbf{S} \in \mathbb{R}^{L \times 4 \times 3}$.

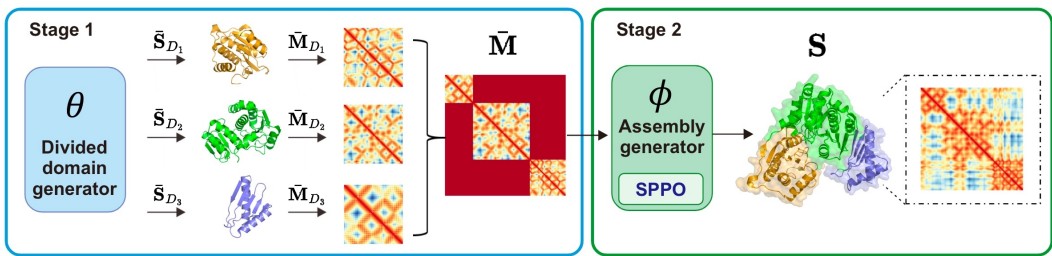

Figure 2: ProteinWeaver employs a two-staged 'divide-and-Assembly' framework, first generating individual protein domains and then using an SE(3) diffusion model to flexibly assemble these domains. $\bar{\mathbf{S}}$ represents isolated domains undergoing internal structural modifications for assembly into integrated backbones.

**Overall architecture.** As shown in Fig.2, we can decouple the generation process of backbones into two stages: (1) divided domain generation and (2) domain assembly generation. This decoupling strategy enables us to break down the modeling of complex backbone structures into two manageable steps: generation of individual domains and assembly of these domains, which significantly simplifies the overall modeling process for complex protein structures.

It should be noted that the domain structure $\bar{\mathbf{S}}$ generated in stage 1 will change when assembled into the final structure $\mathbf{S}$. In other words, we are considering the flexible assembling of domains.

**Divided domain generation.** Given the target protein backbone's length $L$, the structure can be divided into domains $D = \{D_1, D_2, ..., D_m\} \in \text{partition}([1, 2, ...L], m)$ where $\text{partition}(\cdot, m)$ is the function that returns all sequence partition[1] dividing the input sequence into $m$ parts, such that $D_i \cap D_j = \emptyset$ and $\bigcup_{i=1}^{m} D_i = \{1, 2, ..., L\}$. In practice, we can artificially construct $\{D_1, D_2, ..., D_m\}$ where the length of each domain $D_i$ is $L_i$. Given the high designability of existing backbone generation methods for short sequences (Watson et al., 2023; Ingraham et al., 2023; Yim et al., 2023; 2024; Bose et al., 2023; Lin & AlQuraishi, 2023; Lee et al., 2022; Wu et al., 2024a), we directly apply these established models to sample and generate individual domains. Given the domain length $L_i$, $f_\theta : \mathbb{N}^+ \to \mathbb{R}^{L_i \times 4 \times 3}$ generates individual domains $\bar{\mathbf{S}}_{D_i}$. Here, $\theta$ represents domain generator module, $\bar{\mathbf{S}} \in \mathbb{R}^{L \times 4 \times 3}$ is domain residue coordinates before assembly and $\bar{\mathbf{S}}_{D_i}$ is the element corresponding to the index of $D_i$. Various backbone generation methods such as FrameDiff, Chroma, and RFdiffusion can be applied here to generate individual domains.

**Domain assembly generation.** We focus on designing the assembled structure given independently generated domains $\{\bar{\mathbf{S}}_{D_1}, \bar{\mathbf{S}}_{D_2}, ..., \bar{\mathbf{S}}_{D_m}\}$ (Fig.2). We assume the structure of individual domains can undergo overall spatial rotation and translation, along with inner-domain structural perturbations, to generate variable $\mathbf{S}$. To complete this flexible assembling process, we constructed a diffusion model conditioned on the spliced distance map of generated domains.

We initialize the process by extracting $C_\alpha$ distance maps $\{\bar{\mathbf{M}}_{D_1}, \bar{\mathbf{M}}_{D_2}, ..., \bar{\mathbf{M}}_{D_m}\}$ from domain backbones $\{\bar{\mathbf{S}}_{D_1}, \bar{\mathbf{S}}_{D_2}, ..., \bar{\mathbf{S}}_{D_m}\}$ derived from stage 1 generation. Then, we obtain the spliced distance map $\bar{\mathbf{M}}$ by diagonal concatenating distance maps $\{\bar{\mathbf{M}}_{D_1}, \bar{\mathbf{M}}_{D_2}, ..., \bar{\mathbf{M}}_{D_m}\}$ using Eq. (1) (SDM$(\cdot)^2$ represents the splicing distance map), and setting non-diagonal portions (representing inter-domain interactions) to -1 as shown in Fig.2.

$$\bar{\mathbf{M}} = \text{SDM}(\bar{\mathbf{M}}_{D_1}, \bar{\mathbf{M}}_{D_2}, ..., \bar{\mathbf{M}}_{D_m}) = \begin{bmatrix} \bar{\mathbf{M}}_{D_1} & -\mathbf{1} & \cdots & -\mathbf{1} \\ -\mathbf{1} & \bar{\mathbf{M}}_{D_2} & \ddots & \vdots \\ \vdots & \ddots & \ddots & -\mathbf{1} \\ -\mathbf{1} & \cdots & -\mathbf{1} & \bar{\mathbf{M}}_{D_m} \end{bmatrix}. \tag{1}$$

This spliced distance map $\bar{\mathbf{M}}$ initializes the edge representation, providing condition for the diffusion model, as shown in Appendix B.2.1. To enhance the spatial flexibility of domain assembly, we insert a 15-residue linker between adjacent domains and set its interaction with other resions to $-\mathbf{1}$. The model is tasked with learning the deformations generated by each domain in $\{\bar{\mathbf{S}}_{D_1}, \bar{\mathbf{S}}_{D_2}, ..., \bar{\mathbf{S}}_{D_m}\}$ during the combination process as well as the patterns of the ultimate interactions.

To assemble generated domains, we utilize FrameDiff (Yim et al., 2023), the SE(3) diffusion model to pretrain the domain assembly model. As illustrated in the Appendix B.2.1, the assembly module contains a series of folding blocks, where each folding block processes node representation, edge representation and structural frames.

Given the spliced distance map $\bar{\mathbf{M}}$, we iteratively sample the assembled backbone structure through

$$(\hat{\mathbf{T}}^{(0)}, \hat{\psi}) = g_\phi(\mathbf{T}^{(t)}, t, \bar{\mathbf{M}}), \tag{2}$$

where $g_\phi : \text{SE}(3)^L \times [0, 1] \times \mathbb{R}^{L \times L} \to \text{SE}(3)^L \times \mathbb{R}$, $t$ is the diffusion step, $\mathbf{T}^{(t)}$ is the iteratively sampled rigid bodies, $\hat{\mathbf{T}}^{(0)}$ is the predicted assembled backbone structure's rigid bodies and $\hat{\psi}$ is the predicted torsion angle. When the diffusion process ends at time $t_0$, we can obtain the backbone coordinates $\mathbf{S}$ based on $[\mathrm{N}^*, \mathrm{C}_\alpha^*, \mathrm{C}^*, \mathrm{O}^*]$ by applying $\hat{\mathbf{T}}^{(0)}$ and rotation angle $\hat{\psi}$.

---

[1]a 'sequence partition' refers to the process of dividing a sequence into several continues sub-sequences, such as partition$([1, 2, 3, 4], 2) = \{[[1, 2, 3], [4]], [[1, 2], [3, 4]], [[1], [2, 3, 4]]\}$

[2]In this paper, SDM$(\bar{\mathbf{M}}_{D_1}, \bar{\mathbf{M}}_{D_2}, ..., \bar{\mathbf{M}}_{D_m})$, SDM$(\bar{\mathbf{S}}_{D_1}, \bar{\mathbf{S}}_{D_2}, ..., \bar{\mathbf{S}}_{D_m})$, SDM$(\mathbf{S})$ all represent the diagonal concatenation of the distance maps corresponding to the refolded domains of a given structure $\mathbf{S}$.

## 2.2 TRAINING

In this section, we provide the details of ProteinWeaver training, including datasets, pretraining, and preference alignments. More details can be found in Appendix B.2.1.

**Dataset.** To train ProteinWeaver, we constructed a set composed of $(\bar{\mathbf{M}}, \mathbf{S})$ pairs. The structure of protein backbone $\mathbf{S}$ in this set is sourced from the Protein Data Bank (PDB). Following Yim et al. (2023), we filter for single-chain monomers between length 60 and 512 with resolution $< 5$Å downloaded from PDB (Berman et al., 2000) on March 2, 2024. We further filtered the data with more than $50\%$ loops and left with 22728 proteins. Multi-domain PDB structures were further filtered, resulting 5835 PDB structures for pretraining. For each PDB, we identify the domain number $m$ and domain index $\{D_1, D_2, ..., D_m\}$ through Unidoc (Ribeiro et al., 2019). Finally, we convert them to $C_\alpha$ distance maps and use Eq. (1) to get $\bar{\mathbf{M}}$.

**Pretraining.** In our pipeline, domain assembly generation involves the flexible assembling of domains into integrated backbones. The intra-domain structures are alternated during the assembly for optimal inter-domain interaction. To maintain consistency between the pretraining and inference stages, we refolded each domain $\{\mathbf{S}_{D_1}, \mathbf{S}_{D_2}, ..., \mathbf{S}_{D_m}\}$ to $\{\bar{\mathbf{S}}_{D_1}, \bar{\mathbf{S}}_{D_2}, ..., \bar{\mathbf{S}}_{D_m}\}$ of multi-domain PDB using ESMFold (Lin et al., 2023), mimicking their unassembled states for training.

In the pretraining stage, we adopted the training loss from FrameDiff, comprising two main components: diffusion score-matching loss for translation and rotation, along with auxiliary losses related to the coordinate and pairwise distance loss on backbone atoms, as depicted in Eq. (3).

$$\mathcal{L} = \mathcal{L}_{\text{trans}} + 0.5 \cdot \mathcal{L}_{\text{rot}} + 0.25 \cdot \mathcal{L}_{\text{atom}}^{t<0.25} + 0.25 \cdot \mathcal{L}_{\text{pairwise}}^{t<0.25}. \tag{3}$$

Here, $\mathcal{L}_{\text{trans}}$ computes the loss between predicted translations with reference translations, while $\mathcal{L}_{\text{rot}}$ calculates the loss of rotation scores. $\mathcal{L}_{\text{atom}}$ represents the atom coordinate loss and $\mathcal{L}_{\text{pairwise}}$ computes the pairwise distance loss between predicted and reference positions. Following FrameDiff (Yim et al., 2023), we apply auxiliary losses when $t \leq 0.25$. More details can be found in Appendix B.2.2.

**Preference alignment.** By utilizing Eq. (3) to train the model, ProteinWeaver has acquired knowledge of assembling patterns found in natural proteins. However, during the pretraining stage, Eq. (3) only enables the model to maximine the likelihood of the data set $p(\mathbf{S})$, whereas our target is to maximize the quality of the generated structure $\mathbf{S}$ measured by scTM score. Different assembling approaches can result in varying qualities of flexible assembling structures.

To enable the model to generate proteins with higher scTM scores based on pretraining, we employed Eq. (4) as the alignment objective. Here, $r(\mathbf{S}, \mathbf{S}_{\text{ref}})$ serves as a reward function to provide feedback on the scTM score of $\mathbf{S}$, $\pi_{\text{ref}}$ is a copy of $\pi_\phi$ and remains frozen during alignment, $\beta$ is self-paced learning rate. By adjusting $\beta$, we can maximize $r(\mathbf{S}, \mathbf{S}_{\text{ref}})$, i.e., the scTM score, while ensuring minimal difference between $\pi_\phi$ and $\pi_{\text{ref}}$. We use $\Omega$ to represent all protein backbone structures in the alignment stage.

$$\max_{\pi_\phi} \mathbb{E}_{\mathbf{S}_{\text{ref}} \sim \Omega, \bar{\mathbf{M}}=\text{SDM}(\mathbf{S}_{\text{ref}}), \mathbf{S} \sim \pi_\phi(\mathbf{S}|\bar{\mathbf{M}})} [r(\mathbf{S}, \mathbf{S}_{\text{ref}})] - \beta \mathbb{D}_{\text{KL}}[\pi_\phi(\mathbf{S}|\bar{\mathbf{M}}) || \pi_{\text{ref}}(\mathbf{S}|\bar{\mathbf{M}})]. \tag{4}$$

In particular, we use SPPO for preference alignment(Wu et al., 2024b). To apply this alignment, we need to build the preference dataset and construct the "winning" and "losing" pair. We used Unidoc to split the proteins in PDB into single domain structures, and used TMalign (Zhang & Skolnick, 2005) to deduplicate all domain structures, retaining 100 domain structures. Then, we transform these 100 domain structures into $C_\alpha$ distance maps. Using Eq. (1), we combined these 100 domain structure distance maps to construct 10,000 different spliced distance maps. For each spliced distance map, we use ProteinWeaver to generate 3 structures, and use scTM score to identify winner $\mathbf{S}_w$ and loser $\mathbf{S}_l$ to construct data pairs used for SPPO alignment. Finally, we constructed 10,000 data pairs for SPPO alignment.

$$\mathcal{L}_{\text{SPPO}}(\bar{\mathbf{M}}, \mathbf{S}_w, \mathbf{S}_l; \pi_\phi, \pi_{\text{ref}}, \beta) := \left(\beta \log \frac{\pi_\phi(\mathbf{S}_w|\bar{\mathbf{M}})}{\pi_{\text{ref}}(\mathbf{S}_w|\bar{\mathbf{M}})} - \frac{1}{2}\right)^2 + \left(\beta \log \frac{\pi_\phi(\mathbf{S}_l|\bar{\mathbf{M}})}{\pi_{\text{ref}}(\mathbf{S}_l|\bar{\mathbf{M}})} + \frac{1}{2}\right)^2. \tag{5}$$

We determine the preference according to quality of assembled backbones using scTM metrics. Winner is defined as the data with higher backbone quality $\mathbf{S}_w$ (winner data), along with its corresponding lower quality data $\mathbf{S}_l$ (loser data), which are determined by scTM score. These pair data generated under the same conditions ($\bar{\mathbf{M}}$ in our setting). Our approach finalizes the fine-tuning process by maximizing the loss function $\mathcal{L}_{\text{SPPO}}(\bar{\mathbf{M}}, \mathbf{S}_w, \mathbf{S}_l; \pi_\phi, \pi_{\text{ref}})$, enabling the model to produce a structure that closely resembles the winner data and significantly differs from the loser data within the constraints set by the distance map $\bar{\mathbf{M}}$.

### 2.3 INFERENCE

The overall sampling process is summarized in Algorithm 1. We first sample domain division from $\text{partition}([1, 2, ..., L], m)$. For each domain, we use $f_\theta$ generates corresponding domain. Then, we convert them into $C_\alpha$ distance map and use $\text{SDM}(\cdot)$ to get spliced distance map. Finally, using $\text{SE}(3)$ SDE introduced in Yim et al. (2023), we obtain the rigid frames $\mathbf{T}^{(t_0)}$ and calculate final complete protein backbone $\mathbf{S}$.

---

**Algorithm 1** ProteinWeaver Model Inference

**Require:** domain module $\theta$, assembly module $\phi$, residue numbers $L$, diffusion steps $N_{\text{steps}}$, domain numbers $m$, step interval $\zeta$, stop time $t_0$
  # *division of domains*
  $[D_1, D_2, ..., D_m] \sim \text{partition}([1, 2, ..., L], m)$
  # *domain backbones generation*
  **for** $i \in [1, 2, ..., m]$ **do**
    $\bar{\mathbf{S}}_{D_i} = f_\theta(\text{length}(D_i))$
  **end for**
  # *splicing distance maps*
  $\bar{\mathbf{M}} = \text{SDM}(\bar{\mathbf{S}}_{D_1}, \bar{\mathbf{S}}_{D_2}, ..., \bar{\mathbf{S}}_{D_m})$
  # *protein backbone generation*
  $\gamma = (1 - t_0)/N_{\text{steps}}$
  **for** $i \in [1, 2, ..., L]$ **do**
    $x_i^{(1)} \sim \mathcal{N}(0, \text{Id}_3), r_i^{(1)} \sim \mathcal{N}(0, \text{Id})$
    $\mathbf{T}_i^{(1)} = (x_i^{(1)}, r_i^{(1)})$
  **end for**
  **for** $t = 1, 1 - \zeta, 1 - 2\zeta, ..., t_0$ **do**
    $\hat{\mathbf{T}}^{(0)} = g_\phi(\mathbf{T}^{(t)}, t, \bar{\mathbf{M}})$
    $\{(s_n^r, s_n^x)\}_{n=1}^L = \nabla_{\mathbf{T}^{(t)}} \log p_{t|0}(\mathbf{T}^{(t)}|\hat{\mathbf{T}}^{(0)})$
    $\mathbf{T}^{(t-\zeta)} = \text{SDE}_{(\text{SE3})}(\mathbf{T}^{(t)}, \{(s_n^r, s_n^x)\}_{n=1}^L)$
  **end for**
  # *calculate the coordinates*
  $\mathbf{S} = \text{CALC\_COORDINATE}(\mathbf{T}^{(t_0)})$
  **return** $\mathbf{S}$

---

## 3 EXPERIMENTS

We evaluate ProteinWeaver against state-of-the-art protein backbone generation strategies, demonstrating its superiority in four key areas: (1) Domain assembly: ProteinWeaver generates high-quality domain-assembled backbones across diverse domain sources in Sec.3.2. (2) General backbone design: ProteinWeaver outperforms existing methods like RFdiffusion in creating novel, high-quality long-chain backbones in Sec.3.3. (3) Cooperative function design: ProteinWeaver generates function-cooperated backbones through targeted domain assembly, as evidenced by case studies in Sec.3.4. (4) Ablation study in Sec.3.5.

### 3.1 EXPERIMENTAL SETUPS

**Tasks.** We evaluate ProteinWeaver's performance using two tasks: (1) Domain assembly: this is a new protein backbone design task, introduced in our study, lacking existing deep learning-based baselines. It serves both as an appropriate scenario to evaluate our approach and as an important protein engineering task previously studied using traditional method (Huddy et al., 2024). (2) Backbone design: This task focuses on the design of de novo proteins, a subject of significant interest in the global research community. When generating backbone, we use the best of 3 operation (bo3). More details are provided in Appendix B.6.

**Baselines.** We compare ProteinWeaver with various representative protein backbone design baselines, including Chroma (Ingraham et al., 2023), RFdiffusion (Watson et al., 2023), and FrameFlow Yim et al. (2024). To verify the effectiveness of alignment, we also compare a supervised fine-tuning (SFT) based fine-tuning strategy.

**Dataset.** We pretrained the diffusion model for domain assembly using 5,835 filtered multi-domain PDBs from the RCSB database (Berman et al., 2000). For the preference alignment, we performed pairwise combinations of 100 single-domain structures, generating 10,000 pairs to build winner and loser datasets for alignment. Detailed methodology is provided in Appendix B.1.1 and B.1.2.

**Metrics.** We evaluate generation conformations as to their overall backbone *quality, interface quality, novelty, and diversity*. We also provide interface scTM and scRMSD metrics for domain assembly quality evaluation. More details are provided in Appendix B.5.

## 3.2 Evaluation of Protein Domain Assembly

**ProteinWeaver demonstrates strong capability in assembling native domains.** We assessed ProteinWeaver's domain assembly capacity using split domains from native PDBs. As shown in Fig.3A and Tab.1, ProteinWeaver effectively assembles PDB domains (red bars) to form high-quality integrated backbones (mean scTM of 0.88 and mean scRMSD of 2.15). The inter-domain scTM evaluation displays high mean scTM scores of 0.74, suggesting highly consistent domain interfaces between the designed backbone and the ESMfold-refolded structure. Notably, the backbone structures of domains undergo significant alterations after the assembly process, highlighting ProteinWeaver's ability to integrate domains flexibly (Fig.10). To further challenge ProteinWeaver, we used randomly sampled domains from CATH (Orengo et al., 1997) with uncertain assemblability. This resulted in decreased performance (pink bars in Fig. 3A), as expected given the increased difficulty of the task. Despite the performance decrease, we include these results as a benchmark for this challenging task to benefit future studies.

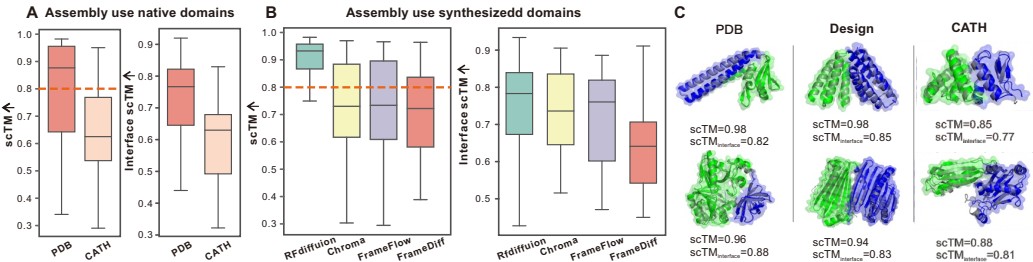

Figure 3: ProteinWeaver enables high-quality backbone design by assembling domains from diverse sources. **(A)** Backbone and interface quality estimation of native domain assembly. **(B)** Backbone and interface quality estimation of synthesized domain assembly. **(C)** Case studies showing the diverse assembled domains. The designed backbone and the refolded backbones (grey) are aligned with assembled backbones (green and blue color coding to different domains). The evaluation was conducted without employing the best-of-three filter.

**ProteinWeaver exhibits strong generalizability in assembling synthesized domains.** To evaluate ProteinWeaver's robustness in assembling distinct domains, we conducted tests using domains synthesized by RFdiffusion. As shown in Fig.3B and Tab.2, ProteinWeaver effectively assembled these synthesized domains into high-quality backbones (green bar), achieving median scTM scores of 0.92, comparable to results with native split PDB domains. We also observed satisfactory interdomain interface quality, with an mean interface scTM of 0.80. We further assessed ProteinWeaver's assembly capacity using domains generated by various backbone design approaches. While domains from other methods could also be assembled, results showed decreased performance for Chroma (yellow), FrameFlow (purple), and FrameDiff (red). This observation may suggest the quality of individual domains impacts the overall quality of assembled backbones. Case studies presented in Fig.3C demonstrate ProteinWeaver's generalizability in assembling domains from different sources, with diverse topologies and secondary structures, into high-quality integrated backbones.

## 3.3 Evaluation of Protein Backbone design

**ProteinWeaver significantly improves the designability of long-chain backbones.** ProteinWeaver facilitates protein backbone design through the assembly of synthesized domains. We evaluated the performance against state-of-the-art methods across protein lengths ranging from 100 to 800 residues (Fig.4 and Tab.2). For proteins between 100 and 400 residues, all the methods are capable of generating high-quality backbones. However, when it comes to generating proteins with 500 to 800 amino acids, the quality of the baseline methods drops rapidly. In contrast, ProteinWeaver consistently maintains its design ability. Specifically, ProteinWeaver (red line) achieves mean scTM

scores of 0.86 for 500-residue proteins (compared to RFdiffusion's 0.76), 0.79 vs. 0.66 for 600-residue proteins, and 0.68 vs. 0.49 for 800-residue proteins. This represents approximately 13% and 39% performance improvements over RFdiffusion in long-chain backbone design for 500 and 800 residues, respectively. These results underscore the effectiveness of the 'divide-and-assembly' strategy for long-chain backbone design, demonstrating ProteinWeaver's superior performance in designing extended protein structures.

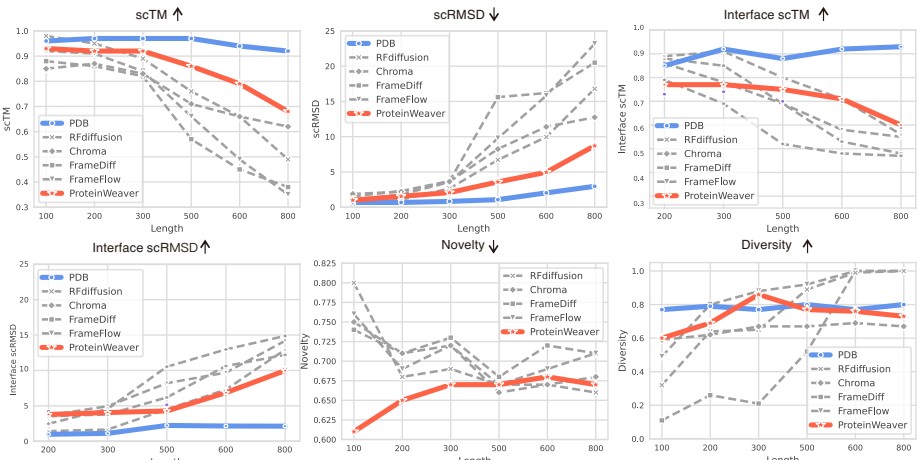

Figure 4: ProteinWeaver shows strong capacity in designing novel and high-quality backbones with significant improvement, particularly in long-chain structures.

**ProteinWeaver significantly enhances the novelty of backbone design.** As shown in Fig.4, ProteinWeaver consistently achieves better novelty compared to existing backbone design approaches for chain lengths ranging from 100 to 300 residues. The advancement may stem from ProteinWeaver's 'divide-and-assemble' strategy, which enables the combination of diverse domains to create novel backbones. For longer chains (500 to 800), other backbone design methods show improved novelty alongside decreased design quality. In contrast, ProteinWeaver generates novel long-chain structures maintaining high quality, demonstrating its strength in designing long protein backbones.

## 3.4 FUNCTION-COOPERATED BACKBONE DESIGN

**ProteinWeaver supports targeted domain assembly, facilitating function-cooperated backbone design.**

Inspired by nature's domain assembly strategy for optimizing and creating new functions (Pawson & Nash, 2003), we applied Protein-Weaver to assemble four protein types, focusing on three functional classes: (1) Nanobodies: simplified antibodies that defend against diseases (De Meyer et al., 2014); (2) Scaffold proteins: target-directed "GPS" molecules (Burack & Shaw, 2000); and (3) Enzymes: proteins that catalyze chemical reactions (Kirk et al., 2002). As shown in Fig.5, ProteinWeaver efficiently assembles these proteins, generating

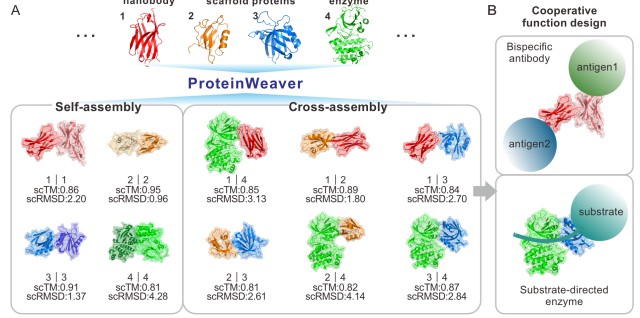

Figure 5: Case studies showing ProteinWeaver potentially enables cooperative function design through the assembly of assigned proteins.

stable interfaces in both self-assembly and cross-assembly configurations. The resulting backbones present potential function-cooperative designs worthy of further investigation. For instance, assembled nanobodies could simultaneously bind different antigens, potentially enhancing synergistic effects (Lewis et al., 2014), while scaffold-enzyme assemblies may improve enzyme target selec-

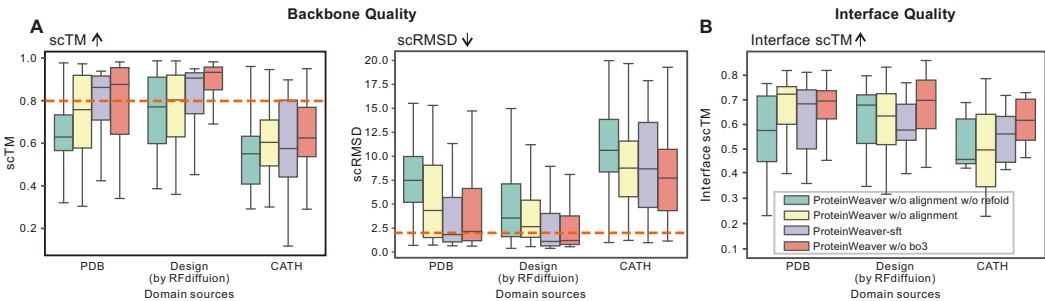

Figure 6: Ablation study on domain assembly. **(A)** The backbone quality and **(B)** Interface quality are evaluated. Distinct sourced-domains are tested. The evaluation was conducted without employing the best-of-three filter.

tivity (Park et al., 2023) (Fig.5). These cases provide biologically significant proof-of-concept, demonstrating ProteinWeaver's ability to create a vast space for functional optimization and design.

### 3.5 ABLATION STUDY

**Training using refolded domains is crucial for ProteinWeaver to learn domain assembly.** We hypothesized that training using refolded domains, which mimics their isolated states, facilitates ProteinWeaver's learning of structural interactions during domain assembly. To evaluate this, we compared ProteinWeaver trained on refolded domains (yellow bars) with a version trained on domains directly split from PDBs without refolding (green bars). As shown in Fig.6 and Tab.1, the model trained on directly split structures exhibited significantly impaired performance in domain assembly tasks. This phenomenon was consistently observed in both domain assembly and protein backbone design tasks (Fig.7). These results demonstrate the importance of using refolded domain structures in model training for effective domain assembly learning.

**Preference alignment efficiently optimizes performance for domain assembly and backbone design.** To evaluate the effectiveness of preference alignment in optimizing domain assembly, we compared ProteinWeaver with alignment (red bar) to a version without alignment (yellow bar). As shown in Fig.6 and Tab.3, alignment significantly improved performance across backbones assembled using various domains, including split domains from PDB, designed domains, and CATH domains. Inter-domain scTM evaluation further confirmed these results, demonstrating preference alignment's high efficiency in optimizing domain interaction quality. We extended this ablation study to general backbone design (Fig.7). ProteinWeaver with alignment (purple line) consistently outperformed the version without alignment (blue line) in generating assembled domains of higher quality. This suggests that alignment is effective in optimizing inter-domain interactions across different design tasks.

**Preference alignment outperforms Supervised Fine-tuning in optimizing domain assembly.** We conducted ablation studies to compare the effectiveness of preference alignment with Supervised Fine-tuning (SFT), an alternative method for optimizing high-quality domain assembly. For SFT, we utilized the "winner" dataset from the preference alignment process to fine-tune the model. As shown in Fig.7, ProteinWeaver fine-tuned with preference alignment (red bar) significantly outperformed the version fine-tuned with SFT (purple bar). These results strongly suggest that preference alignment is more effective than SFT for optimizing domain assembly in this task.

## 4 RELATED WORK

### 4.1 DIFFUSION MODELING ON PROTEIN BACKBONE

Inspired by the considerable success of diffusion models across various fields (Ho et al., 2020; Song et al., 2020), researchers have applied this approach to protein structure and sequence design. Anand et al. pioneered this effort with a co-diffusion model for backbone, sequence, and sidechain generation using AlphaFold2's Structure Module (Anand & Achim, 2022) . Subsequent methods explored diffusion on inter-residue geometry (Lee et al., 2023) and backbone dihedral angles (Wu et al.,

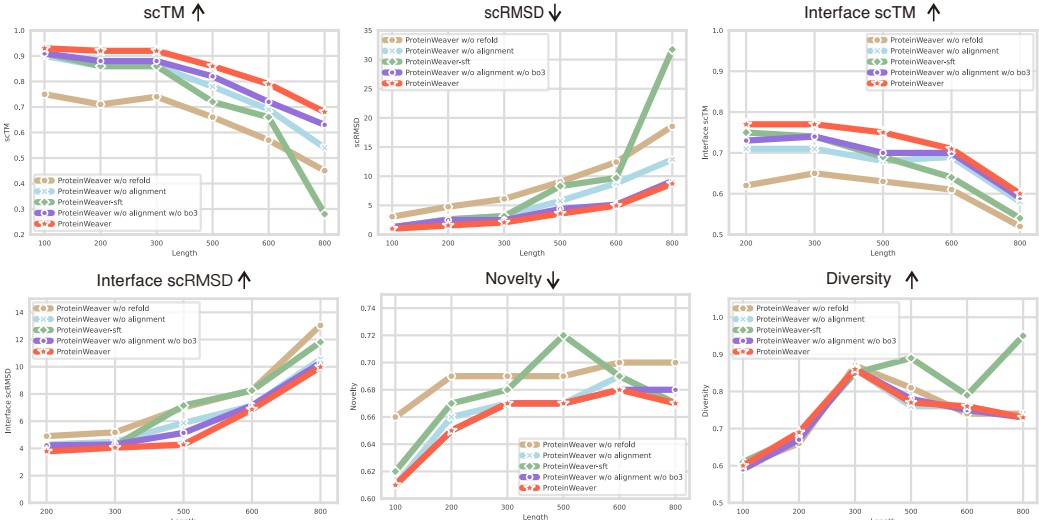

Figure 7: Ablation study on backbone design. "bo3" is abbreviation for best of 3.

2024a). Current protein structure diffusion models primarily focus on end-to-end structure generation in SE3 or R3 space (Yim et al., 2023), with extensions to function motif scaffolding (Trippe et al., 2022; Yim et al., 2024). Chroma achieved more general conditioning and improved efficiency through geometrically constrained harmonic constraints (Ingraham et al., 2023), while RFdiffusion demonstrated state-of-the-art designability with experimental validation (Watson et al., 2023). Proteus (Wang et al., 2024) employs AlphaFold2's graph-based triangle approach and multi-track interaction networks. It outperforms RFdiffusion in long-chain monomer generation and exceeds Chroma in complex structure generation. Also, sequence-structure co-design methods, such as MultiFlow (Campbell et al., 2024) and CarbonNovo (Ren et al.) demonstrates superior designability compared to RFdiffusion.

## 5 CONCLUSIONS AND DISCUSSIONS

**Conclusions.** In this paper, we introduce ProteinWeaver, the first protein backbone design model based on a 'divide-and-assembly' framework. This approach enables the creation of novel, high-quality backbones through the flexible assembly of diverse domains. ProteinWeaver demonstrates superior performance in general backbone design, particularly excelling in the challenging task of designing long-chain backbones compared to existing state-of-the-art approaches. The assembly capability of ProteinWeaver not only advances current protein design methods but also lays the groundwork for cooperative function design, potentially opening new avenues for protein engineering.

**Discussions.** While our framework demonstrates a new paradigm for protein backbone design, we acknowledge several technical aspects that warrant further investigation: (1) Recent advancements have revealed that flow matching-based protein backbone generation methods outperform diffusion-based approaches. An exciting avenue for future research lies in exploring whether replacing our current diffusion-based framework with a flow matching-based approach could yield further benefits; (2) Domain structure representation: Our current method explicitly represents local domain structures as distance maps. However, the efficacy of alternative pairwise representation methods remains an open question. (3) Integration of design stages: Currently, our approach employs two separate steps for backbone design. A promising direction for future research is developing an end-to-end unified framework that efficiently integrates these steps. Future research could further refine and extend this framework by addressing these questions, potentially revolutionizing our ability to create functional protein backbones with unprecedented precision.

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

## A    BRIEF INTRODUCTION OF DOMAIN ASSEMBLY

Domain assembly is a well-established task in protein design that has been extensively researched using traditional computational biology methods, such as RFdiffusion. To clarify, we provide the following examples that highlight the significance of this task:

**[Domain-Assembly for Structural Design]** A recent study demonstrated the use of helical protein blocks for designing extendable nanomaterials (Huddy et al., 2024). This work serves as a proof of concept for the domain-based design approach, but it focused solely on pure helical-bundle assemblies. In contrast, our study generalizes this idea to encompass a broader range of domains with structural variations. We present the first application of deep learning methods for general domain assembly, marking an important new paradigm in this field.

[**Domain assembly for functional design**] Nearly two decades ago, research showed that well-assembled domains, through interface-directed evolution, could dramatically enhance both affinity and specificity—achieving over 500-fold and 2,000-fold increases, respectively (Huang et al., 2008). More recently, another study revealed that assembling a substrate recruitment domain to an enzyme significantly improved its functional specificity (Park et al., 2023). These studies underscore the validity and importance of our motivation for pursuing domain-assembly-based protein structure and function design, a concept we introduced in our manuscript. We have added more references in the revised manuscript to emphasize this importance.

## B    EXPERIMENTAL SETUP

### B.1    DATA PROCESSING

#### B.1.1    PRETRAINING DATA

To train ProteinWeaver, specifically the assembly generator, a series of spliced distance maps $\bar{\mathbf{M}}$ and the corresponding structures $\mathbf{S}$ are required. In pretraining stage, our structures are sourced from the PDB dataset. Following Yim et al. (2023), we filter for single-chain monomers between length 60 and 512 with resolution $< 5$Å downloaded from PDB (Berman et al., 2000) on March 2, 2024, resulting in 22,728 proteins.

For each protein $\mathbf{S}$, we used Unidoc to identify the domain number $m$ and domain indices $\{D_1, D_2, ..., D_m\}$. For proteins where $m = 1$, , we directly converted their structures into the corresponding $C_\alpha$ distance map $\bar{\mathbf{M}}$. If $m > 1$, we initially segmented the PDB structure into domain structures $\{\mathbf{S}_{D_1}, \mathbf{S}_{D_2}, ...\mathbf{S}_{D_m}\}$ based on the domain indices identified by Unidoc. Subsequently, we refolded each domain structure using ESMFold to construct the refolded domain structures $\{\bar{\mathbf{S}}_{D_1}, \bar{\mathbf{S}}_{D_2}, ...\bar{\mathbf{S}}_{D_m}\}$. Refolding with ESMFold ensures the independence of domain structures, maintaining consistency between training and inference.

Finally, we converted the refolded domain structures $\{\bar{\mathbf{S}}_{D_1}, \bar{\mathbf{S}}_{D_2}, ...\bar{\mathbf{S}}_{D_m}\}$ into their corresponding $C_\alpha$ distance map $\{\bar{\mathbf{M}}_{D_1}, \bar{\mathbf{M}}_{D_2}, ..., \bar{\mathbf{M}}_{D_m}\}$ and utilized Eq. (1) to transform them into spliced distance map $\bar{\mathbf{M}}$. Among these 22,728 proteins, Unidoc identified 5,835 multi-domain PDB structures.

### B.1.2 ALIGNMENT DATA

In this paper, we utilized SPPO for preference alignment of the pretrained models. To construct the alignment dataset, a series of pair datas consisting of winner data $\mathbf{S}_w$ and loser data $\mathbf{S}_l$ needed to be obtained. To achieve this, the following steps were undertaken: (1) Construction of the distance map $\bar{\mathbf{M}}$. (2) Generation of multiple structural data using $\bar{\mathbf{M}}$ as a condition. (3) Selection of metrics to evaluate the quality of data, leading to the construction of winner data and loser data.

To construct the distance map $\bar{\mathbf{M}}$, similar to building the pretraining dataset, we segmented the proteins from the PDB into single-domain structures using Unidoc. Subsequently, we de-duplicated these single-domain structures using TMalign (with a threshold set at 0.3). Following this, we randomly selected 100 single domains and combined them randomly to create 10,000 domain combinations. Eq. (1) was then applied to derive the corresponding spliced distance map for each combination.

For each spliced distance map, ProteinWeaver was employed to generate three structures, with the scTM score utilized as the metric for distinguishing between winner and loser data. Ultimately, we established 10,000 pairs of pair datas for SPPO alignment.

## B.2 MODEL

### B.2.1 MODEL ARCHITECTURE

ProteinWeaver iteratively updates the structural frames of proteins through a sequence of $L$ layers of folding blocks. A folding block is composed of three modules: an IPA module, a backbone update module, and a edge update module. Each layer of the folding block takes single representation, pair represetation and frames as input.

Following Yim et al. (2023), we add a skip connection and a 2-layers transformer encoder to original IPA (Jumper et al., 2021). The IPA module takes the single and pair representations and frames as input. For edge update module, we use a linear layer operates on pair representation with a single representation-gated structural bias. For backbone update module, we use a MLP to predict translation and rotation updates for the frames of each residue, where the input to MLP is single represetation.

There are no weights shared among the $L$ folding blocks. The sequence representation is initialized using the diffusion timestep and the edge representation is initialized using spliced distance map.

### B.2.2 SE(3) DIFFUSION

**Protein Backbone Diffusion model on SE(3).** In protein backbone design, a common modeling approach involves the utilization of SE(3) diffusion as proposed in Yim et al. (2023). In this SE(3) diffusion, each amino acid is modeled as a frame, with rotation and translation variables diffused separately to complete the modeling process. The forward diffusion formula for this SE(3) diffusion is as follows:

$$d\mathbf{T}^{(t)} = [\mathbf{0}, -\frac{1}{2}\mathbf{X}^{(t)}]dt + [d\mathbf{B}^{(t)}_{\mathrm{SO}(3)}, d\mathbf{B}^{(t)}_{\mathbb{R}^3}]. \tag{6}$$

Here, $\mathbf{B}$ represents Brownian motion ($\mathbf{B}_{\mathrm{SO}(3)}$ and $\mathbf{B}_{\mathbb{R}^3}$ represent Brownian motion on $\mathrm{SO}(3)$ and $\mathbb{R}^3$ respectively), $t \in [0, 1]$. $t = 1$ is initialized noise data and $t = 0$ is final sampling data.

In Yim et al. (2023), the forward diffusion probability of the translation variable is defined as $p_{t|0}(x^{(t)}|x^{(0)}) = \mathcal{N}(x^{(t)}; e^{-t/2}x^{(0)}, (1 - e^{-t})Id_3)$. Based on the forward diffusion probability formula, we can obtain the explicit score of the translation variable using Eq. (7).

$$\nabla \log p_{t|0}(x^{(t)}|x^{(0)}) = (1 - e^{-t})^{-1}(e^{-t/2}x^{(0)} - x^{(t)}). \tag{7}$$

Correspondingly, the forward diffusion probability on $\mathrm{SO}(3)$ is defined as $p_{t|0}(r^{(t)}|r^{(0)}) = f(\omega(r^{(0)^T}r^{(t)}), t)$, where $\omega(r)$ is the rotation angle in radians for any $r \in \mathrm{SO}(3)$. The probability estimation on $\mathrm{SO}(3)$ can be represented as

$$f_{\text{IGSO3}}(\omega, t) = \sum_{\ell \in \mathbb{N}} (2\ell + 1) e^{-\ell(\ell+1)t/2} \frac{\sin\left((\ell + 1/2)\omega\right)}{\sin(\omega/2)}. \tag{8}$$

Based on Eq. (8), we can obtain the score of the rotation variable on $\text{SO}(3)$ as follows:

$$\nabla \log p_{t|0}(r^{(t)}|r^{(0)}) = \frac{r^{(t)}}{\omega(t)} \log(r^{(0,t)}) \frac{\partial_\omega f(\omega(t), t)}{f(\omega(t), t)}. \tag{9}$$

By training based on Eq. (7) and Eq. (9), we can obtain denoising networks on $\text{SO}(3)$ and $\mathbb{R}$, thereby completing the denoising process for $\text{SE}(3)$. Specifically, we generate domains of varying lengths using models like Chroma, RFdiffusion, and FrameFlow. We filter for domains with scTM $>0.8$, then assemble them using ProteinWeaver.

### B.3 TRAINING DETAILS

#### B.3.1 LOSS

Following Yim et al. (2023), we use DSM loss ($\mathcal{L}_{\text{trans}}$ and $\mathcal{L}_{\text{rot}}$) to learn the translation and rotation score to ensure that the model learns the correct $\text{SE}(3)$ diffusion process.

**Auxiliary losses.** We use two additional losses to learn atom distance and directly penalize atomic errors in the final time step of generation. For a generated protein backbone structure $\mathbf{S} \in \mathbb{R}^{L \times}$, we have a referece backbone structure $\hat{\mathbf{S}}$. Correspondingly, we can get the coordinates of all atoms $\mathbf{A} \in \mathbb{R}^{N_{\text{atom}} \times 3}$ corresponding to the generated protein backbone structure and the coordinates of all atoms $\hat{\mathbf{A}}$ of the reference backbone structure. where $N_{\text{atom}}$ is the number of atoms of the generated structure. After that, by calculating the relative distance between atoms, we can get the predicted atomic relative distance matrix $\mathbf{M_A} \in \mathbb{R}^{N_{\text{atom}} \times N_{\text{atom}}}$ and reference atomic relative distance matrix $\hat{\mathbf{M}}_{\hat{\mathbf{A}}} \in \mathbb{R}^{N_{\text{atom}} \times N_{\text{atom}}}$.

The first loss is a direct MSE on the backbone (bb) positions,

$$\mathcal{L}_{\text{atom}} = \frac{1}{N} \|\mathbf{S}^{(0)} - \hat{\mathbf{S}}^{(0)}\|^2, \tag{10}$$

and the second loss is a local neighborhood loss on pairwise atomic distances,

$$\mathcal{L}_{\text{pairwise}} = \frac{1}{N^2} \|\mathbf{M_A}^{(0)} - \hat{\mathbf{M}}_{\hat{\mathbf{A}}}^{(0)}\|^2. \tag{11}$$

#### B.3.2 ADDITIONAL DETAILS

**Hyperparameters.** In training stage, we train the ProteinWeaver with the "time batching" scheme described in Yim et al. (2023). In the "time batching" scheme, each batch contains multiple time steps of a protein from a cluster. We set the weight of auxiliary losses to 0.25 and used a single represetation embedding dimension of 256, an pair represetation demension of 128. The learning rate is set to 0.0001. Moreover, we used ADAM (Kingma, 2014) as our optimizer. In the alignment stage, The learning rate is set to 0.00001 and SPPO beta is set to 1.0.

**Training hardware setup.** ProteinWeaver is coded in PyTorch and was trained on 8 V100 32GB NVIDIA GPUs for 10 days.

### B.4 INFERENCE DETAILS

During the inference process, when constructing the spliced distance map $\bar{\mathbf{M}}$, we introduced a linker of 15 amino acids in length, with the corresponding region in $\bar{\mathbf{M}}$ set to $-1$. We observed a significant enhancement in the flexibility of domain fusion upon the addition of the linker, particularly noticeable in the generation of short proteins. Additionally, during the inference stage, we set $\bar{\mathbf{M}}$ to a matrix with all elements as $-1$ when $t < 0.2$. This adjustment similarly contributes to the quality of protein generation by the model.

## B.5 EVALUATION METRICS

**scTM and scRMSD.** The self consistency template modeling score (scTM) and the self consistency root-mean-square deviation (scRMSD) are two commonly used metrics for quality of protein backbone design.

Template Modeling score (TM-score) is a metric used in structural bioinformatics to assess the similarity between a predicted protein structure and a known target structure. the TM-score between a predicted structure $\mathbf{T}_{\text{predicted}} \in \text{SE}(3)^L$ and a target structure $\mathbf{T}_{\text{target}} \in \text{SE}(3)^L$ defined by

$$\text{TM-score}(\mathbf{T}_{\text{predicted}}, \mathbf{T}_{\text{target}}) = \max \left( \frac{1}{L_{\text{target}}} \sum_{i=1}^{L_{\text{aligned}}} \frac{1}{1 + \left( \frac{d_i}{d_0}(L_{\text{target}}) \right)^2} \right), \quad (12)$$

where $L_{\text{target}}$ is the length of the amino acid sequence of the target protein, and $L_{\text{common}}$ is the number of residues that appear in both the predicted and target structures. $d_i$ is the distance between the $i^{th}$ pair of residues in the predicted and target structures, and $d_0(L_{\text{target}}) = 1.24\sqrt[3]{L_{\text{target}} - 15} - 1.8$ is a distance scale that normalizes distances.

The root-mean-square deviation (RMSD) is a simple metric over paired residues defined by

$$\text{RMSD}(\mathbf{T}_{\text{predicted}}, \mathbf{T}_{\text{target}}) = \sqrt{\frac{1}{L} \sum_{i=1}^{L} d_i^2}, \quad (13)$$

where $d_i^2$ is the distance between atom $i$ and either a reference structure or the mean position of the $L$ equivalent atoms. This is often calculated for the backbone heavy atoms C, N, O, and $C_\alpha$ or sometimes just the $C_\alpha$ atoms.

For a generated protein structure $\mathbf{T}_0 \in \text{SE}(3)^L$, we apply ProteinMPNN (Dauparas et al., 2022) to sample multiple ($N_{\text{seq}}$) sequences. Each sequence is then folded with ESMFold (Lin et al., 2023) to obtain the target backbone $\hat{\mathbf{T}}_0 = [\mathbf{T}_1, \mathbf{T}_2, ..., \mathbf{T}_{N_{\text{seq}}}]$ where $\mathbf{T}_i \in \text{SE}(3)^L$. The scTM score and scRMSD can be expressed for the generated protein structure $\mathbf{T}_0$ and the target backbone $[\mathbf{T}_1, \mathbf{T}_2, ..., \mathbf{T}_{N_{\text{seq}}}]$ as

$$\text{scTM}(\mathbf{T}_0, \hat{\mathbf{T}}_0) = \max \left( \text{TM-score}(\mathbf{T}_0, \mathbf{T}_1), \text{TM-score}(\mathbf{T}_0, \mathbf{T}_2), ..., \text{TM-score}(\mathbf{T}_0, \mathbf{T}_{N_{\text{seq}}}) \right), \quad (14)$$

$$\text{scRMSD}(\mathbf{T}_0, \hat{\mathbf{T}}_0) = \min \left( \text{RMSD}(\mathbf{T}_0, \mathbf{T}_1), \text{RMSD}(\mathbf{T}_0, \mathbf{T}_2), ..., \text{RMSD}(\mathbf{T}_0, \mathbf{T}_{N_{\text{seq}}}) \right). \quad (15)$$

**Interface scTM and Interface scRMSD.** To better assess the rationality of interface design between generated protein domains, we introduced the interface scTM and interface scRMSD metric. For the generated protein $\mathbf{T}_0$, we employed Unidoc to identify and partition its domains, selecting regions where the $C_\alpha$ distance between amino acids within domains was $< 12\text{Å}$ as interface $\mathbf{I}_0$. Subsequently, utilizing ProteinMPNN and ESMFold, we refolded $N_{\text{seq}}$ interface structures $\hat{\mathbf{I}}_0 = [\mathbf{I}_1, \mathbf{I}_2, ..., \mathbf{I}_{N_{\text{seq}}}]$ from $\mathbf{T}_0$ and computed the scTM between $\mathbf{I}_0$ and $\hat{\mathbf{I}}_0$.

$$\text{Interface-scTM}(\mathbf{T}_0, \hat{\mathbf{T}}_0) = \max \left( \text{TM-score}(\mathbf{T}_0, \mathbf{T}_1), ..., \text{TM-score}(\mathbf{T}_0, \mathbf{T}_{N_{\text{seq}}}) \right), \quad (16)$$

$$\text{Interface-scRMSD}(\mathbf{T}_0, \hat{\mathbf{T}}_0) = \min \left( \text{RMSD}(\mathbf{T}_0, \mathbf{T}_1), ..., \text{RMSD}(\mathbf{T}_0, \mathbf{T}_{N_{\text{seq}}}) \right). \quad (17)$$

This approach allowed us to evaluate the quality of the designed interfaces within the generated protein, facilitating a comprehensive analysis of domain splicing-based protein backbone design. In the evaluation conducted in this study, we focused solely on structures with an scTM score $> 0.5$ for the assessment of interface scTM. This approach is particularly advantageous for evaluating the model's capability to interact with domains of higher design quality.

**Max TM.** We calculate novelty using the maximum TM-score of designable generated proteins (scTM score $> 0.7$) to the PDB data. To expedite the computation of metrics, in this study, we

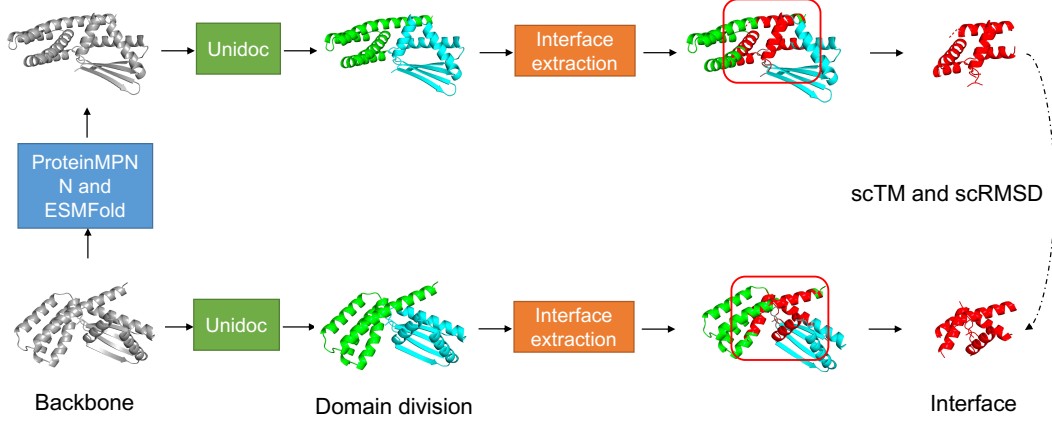

Figure 8: Interface scTM and Interface scRMSD. Similar to computing the scTM score and scRMSD, to assess the quality of the domain-domain interface, we utilized ProteinMPNN and ESM-Fold to refold the given structure, segmented the interface structure, and calculated the scTM score and scRMSD of the interface structure before and after refolding.

employed FoldSeek for protein structure compression and utilized **foldseek easy-search** to calculate TM score between the generated proteins and all proteins in the PDB database. This streamlined approach facilitated efficient assessment and comparison of protein structures, enhancing the speed and accuracy of our analysis in the domain splicing-based protein backbone design research.

**Max Clust.** Proteins are usually gathered into different clusters during training. So for diversity, we use the number of generated clusters with a TM-score threshold of 0.5 of the generated samples as our diversity metric (higher is better). Similar to calculating **Max TM**, we utilized FoldSeek to compute diversity in our study. Specifically, we employed **foldseek easy-cluster** to perform clustering calculations on a set of proteins. This method allowed us to analyze and assess the diversity within protein structures efficiently. Note that in certain model, designability is inversely correlated with diversity as these models can produce unrealistic (e.g. unfolded) proteins that are "diverse" because they do not align well with each other.

### B.6 TASKS

**Protein domain assembly.** In this study, we evaluated the model's capability for domain assembly using domains from three distinct sources: (1) reassembling domains from native PDBs, (2) assembling randomly sampled native domains from CATH domains, and (3) assembling structures generated/synthesized by the backbone design models.

For the experiment involving the reassembly of domains of natural proteins, we curated a test set of 500 proteins from the multi-domain dataset of the Protein Data Bank (PDB). Specifically, we employed Unidoc to identify and partition the domains of these natural proteins, transforming them into spliced distance maps. Following model training, we used these spliced distance maps as conditions to generate corresponding proteins and computed their TM score and scTM score against actual natural proteins. The results indicated that ProteinWeaver has learned the assembly patterns of natural proteins.

In the experiment involving the splicing of CATH domains, we selected 500 proteins with distinct topological structures from the CATH dataset and constructed 10,000 novel splicing combinations using a random assembly approach. Subsequently, we tested these combinations using Protein-Weaver and further use SPPO for preference alignment.

Finally, we also assessed the performance of ProteinWeaver in assembling structures generated by different backbone design models. Notably, in this experiment, we did not filter the generated structures based on conditions. As a result, we observed that RFdiffusion, known for its stable generation quality, demonstrated superior performance across various lengths without conditional screening.

**Protein backbone design.** For protein backbone design, we first generate domains of varying lengths using existing backbone design models like Chroma, RFdiffusion, and FrameFlow. Various models are used to obtain more diverse domain structures. We filter for domains with scTM score >0.8, then assemble them using ProteinWeaver.

In practice, this study only considered the results of assembling two domains during the inference stage. For instance, for a protein of length 500, we can decompose the backbone design into an assembly of a 200-length domain with a 300-length domain or an assembly of 100-length with 400-length. Initially, existing backbone design models were employed to generate fixed-length domains with scTM score >0.8. Subsequently, these two proteins were assembled together to complete the protein backbone design of length 500.

## C  ADDITIONAL RESULTS

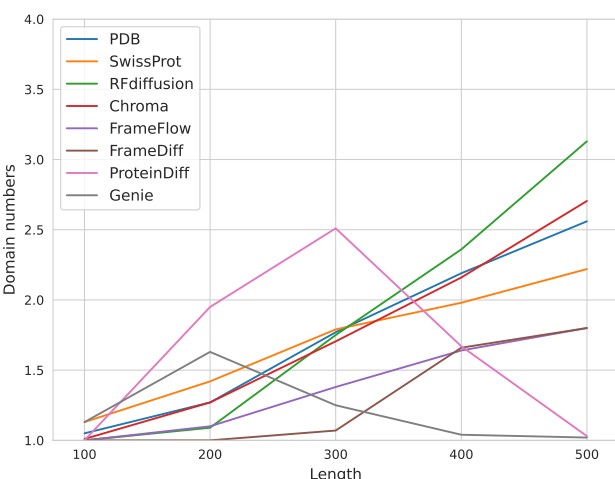

Figure 9: Domain statistics in designed backbone structures. We analyzed the domain composition of backbone structures designed using various methods and compared them to native proteins from RCSB PDB and SwissProt. Our findings reveal distinct trends in domain organization across different protein lengths and design approaches: (1) Native proteins: As protein length increases from 100 to 500 residues, we observe a natural trend of increasing domain numbers, typically ranging from 1 to 3 domains. (2) RFdiffusion and Chroma: These methods closely mimic nature's trend, showing an increase in domain numbers as protein length grows. Other methods (FrameDiff, FrameFlow, ProteinDiff, and Genie): These approaches demonstrate limited capability in generating multi-domain backbones, deviating from the natural trend observed in native proteins. These results highlight the varying abilities of different backbone design methods to capture the complex domain architecture of proteins.

Table 1: Performance of ProteinWeaver on domain assembly using domains derived from native PDB structures and synthetic structures. Intf. quality metrics refer to the domain-domain assembly interface. Intf. is an abbreviation for interface. The reported results are the mean±std of repetitive experiments.

| | Native PDBs | | | | Generated domain | | | | CATH domain | | | |
|---|---|---|---|---|---|---|---|---|---|---|---|---|
| | Backbone Quality | | Intf. Quality | | Backbone Quality | | Intf. Quality | | Backbone Quality | | Intf. Quality | |
| | scTM ↑ | scRMSD ↓ | Intf. scTM ↑ | Intf. scRMSD ↓ | scTM ↑ | scRMSD ↓ | Intf. scTM ↑ | Intf. scRMSD ↓ | scTM ↑ | scRMSD ↓ | Intf. scTM ↑ | Intf. scRMSD ↓ |
| ProteinWeaver w/o alignment w/o refold | 0.63±0.14 | 7.48±3.97 | 0.67±0.15 | 5.00±1.38 | 0.77±0.16 | 3.55±3.78 | 0.73±0.13 | 4.12±0.84 | 0.55±0.15 | 10.62±3.99 | 0.58±0.10 | 5.73±1.26 |
| ProteinWeaver w/o alignment | 0.76±0.19 | 4.34±4.67 | **0.77±0.12** | **4.17±1.10** | 0.80±0.17 | 2.65±3.61 | 0.72±0.12 | **3.81±0.57** | 0.60±0.15 | 8.76±4.19 | 0.60±0.16 | 5.98±2.29 |
| ProteinWeaver-sft | 0.86±0.18 | **1.82±4.20** | 0.76±0.13 | 4.36±1.23 | 0.91±0.17 | **1.11±4.67** | 0.74±0.10 | 4.07±0.88 | 0.58±0.21 | 8.18±9.17 | 0.65±0.11 | 5.16±1.52 |
| ProteinWeaver w/o bo3 | **0.88±0.19** | 2.15±4.30 | 0.74±0.12 | 4.39±1.47 | **0.93±0.08** | 1.20±3.61 | **0.80±0.12** | 4.04±0.86 | **0.63±0.17** | **7.72±4.62** | **0.70±0.13** | **4.68±1.15** |

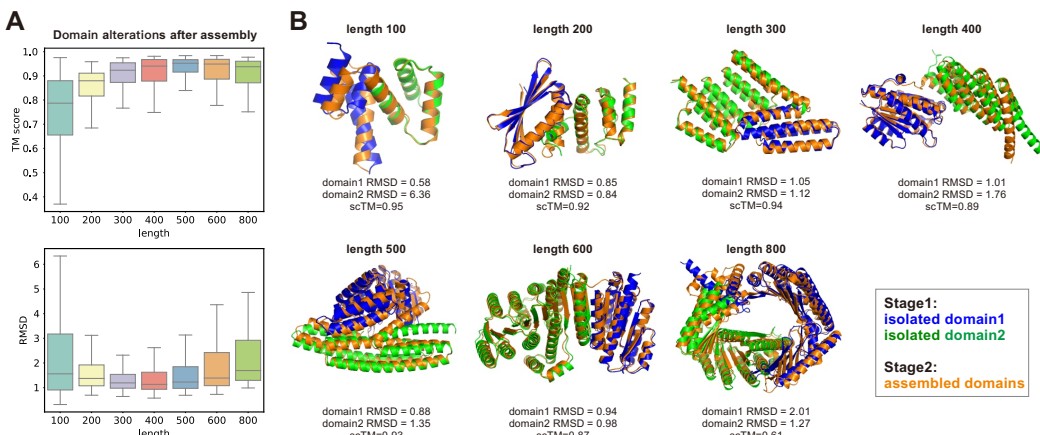

Figure 10: Domains undergo structural alterations after assembly using ProteinWeaver. (A) We analyzed the domain structure alterations between the stage 1 isolated state and the stage 2 assembled state using TM score and RMSD. Significant structural alterations can be observed after assembly. (B) Case study showing the detailed structural alterations. These results highlight ProteinWeaver's capacity in flexible domain assembly.

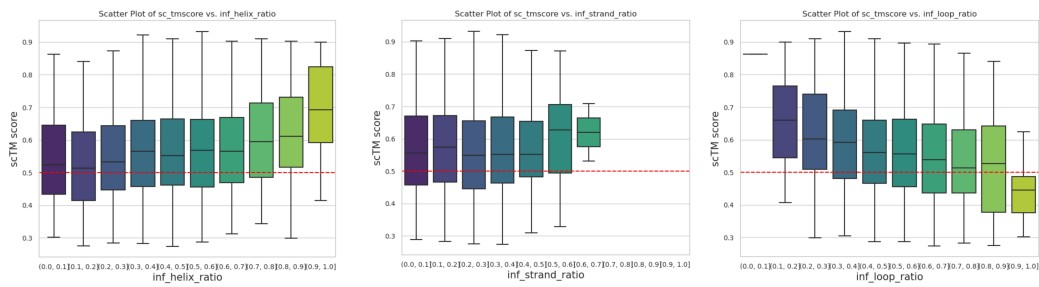

Figure 11: We clustered natural domains in the CATH dataset based on their topological differences, resulting in approximately 530 domain structures that represent a wide distribution of protein domains. This dataset allows us to evaluate the effects of structural variations effectively. We performed pairwise assemblies of these domains and assessed the quality of the designed structures using the scTM score. Our analysis included a comparison of secondary structure ratios at the assembly interface, which we believe directly affects domain assembly.

Table 2: Performance of ProteinWeaver on domain assembly using domains derived from different backbone design models. The performance is evaluated without best of three filter. The reported results are the mean±std of repetitive experiments. We did not report the results of the Interface quality of length 100. This is because these methods only generates one single domain backbones identified by Unidoc. No multi-domain backbones are available for the evaluation. "Intf." is an abbreviation for interface.

| | length 100 | | | | length 200 | | | | length 400 | | | |
|---|---|---|---|---|---|---|---|---|---|---|---|---|
| | Backbone Quality | | Intf. Quality | | Backbone Quality | | Intf. Quality | | Backbone Quality | | Intf. Quality | |
| | scTM ↑ | scRMSD ↓ | Intf. scTM ↑ | Intf. scRMSD ↓ | scTM ↑ | scRMSD ↓ | Intf. scTM ↑ | Intf. scRMSD ↓ | scTM ↑ | scRMSD ↓ | Intf. scTM ↑ | Intf. scRMSD ↓ |
| RFdiffusion | 0.95±0.03 | 0.78±0.03 | – | – | 0.95±0.06 | 1.07±1.07 | 0.81±0.07 | 3.69±0.13 | 0.74±0.17 | 5.41±4.26 | 0.75±0.13 | 4.24±1.13 |
| Chroma | 0.84±0.16 | 1.62±2.45 | – | – | 0.75±0.17 | 4.37±3.29 | 0.75±0.12 | 4.13±0.87 | 0.68±0.13 | 7.45±3.95 | 0.72±0.11 | 4.95±1.21 |
| FrameFlow | 0.84±0.14 | 1.61±1.94 | – | – | 0.75±0.15 | 3.18±2.98 | 0.70±0.13 | 3.84±0.55 | 0.64±0.16 | 8.09±4.31 | 0.73±0.11 | 4.29±0.87 |
| FrameDiff | 0.74±0.17 | 2.68±2.53 | – | – | 0.75±0.13 | 3.44±2.71 | 0.66±0.10 | 3.98±0.52 | 0.64±0.13 | 7.09±3.69 | 0.60±0.10 | 5.11±0.91 |

Table 3: Performance of ProteinWeaver on backbone design models evaluated using various lengths ranging from 100 to 800. For each length, we randomly sampled 50 native PDB structures from RCSB as a golden reference for the task. The reported results are the mean±std of repetitive experiments. When the length is 100, the current backbone design model generates too few multi-domain proteins and is not statistically significant, so it is not reported.

**length100**

| | Backbone Quality | | Intf. Quality | | Novelty | Diversity |
|---|---|---|---|---|---|---|
| | scTM ↑ | scRMSD ↓ | Intf. scTM ↑ | Intf. scRMSD ↓ | Max TM ↓ | Max Clust ↑ |
| Native PDB | 0.96±0.10 | 0.67±1.61 | 0.82±0.17 | 0.75±0.34 | – | 0.77 |
| RFDiffusion | **0.98±0.05** | **0.48±0.56** | – | – | 0.80±0.08 | 0.32 |
| Chroma | 0.85±0.13 | 1.88±1.84 | – | – | 0.75±0.09 | 0.59 |
| FrameFlow | 0.92±0.08 | 1.06±0.94 | – | – | 0.76±0.07 | 0.49 |
| FrameDiff | 0.88±0.08 | 1.54±1.03 | – | – | 0.74±0.07 | 0.11 |
| Proteus | 0.94±0.06 | 0.84±0.52 | – | – | 0.73±0.10 | 0.5 |
| CarbonNovo w/o plm | 0.63±0.09 | 4.26±2.33 | – | – | 0.65±0.06 | 0.94 |
| ProteinWeaver w/o alignment w/o refold | 0.75±0.15 | 3.05±2.28 | 0.64±0.06 | **3.52±0.01** | 0.66±0.05 | 0.60 |
| ProteinWeaver w/o alignment | 0.90±0.14 | 1.26±1.97 | 0.73±0.07 | 3.96±0.71 | **0.61±0.07** | 0.60 |
| ProteinWeaver-sft | 0.91±0.13 | 1.19±1.79 | 0.73±0.07 | 3.59±0.23 | 0.62±0.06 | **0.61** |
| ProteinWeaver w/o bo3 | 0.91±0.14 | 1.27±1.84 | **0.75±0.14** | 3.96±0.78 | **0.61±0.07** | 0.59 |
| ProteinWeaver | 0.93±0.06 | 0.99±0.62 | 0.73±0.10 | 3.71±0.37 | **0.61±0.07** | 0.60 |

**length200**

| | Backbone Quality | | Intf. Quality | | Novelty | Diversity |
|---|---|---|---|---|---|---|
| | scTM ↑ | scRMSD ↓ | Intf. scTM ↑ | Intf. scRMSD ↓ | Max TM ↓ | Max Clust ↑ |
| Native PDB | 0.97±0.08 | 0.67±1.37 | 0.85±0.18 | 0.98±1.15 | – | 0.79 |
| RFDiffusion | **0.95±0.09** | 1.07±1.41 | **0.89±0.13** | **1.43±1.65** | 0.68±0.07 | 0.64 |
| Chroma | 0.87±0.10 | 2.19±1.53 | 0.79±0.17 | 2.47±2.21 | 0.71±0.06 | 0.62 |
| FrameFlow | 0.91±0.11 | 1.79±2.24 | 0.86±0.04 | 3.66±0.26 | 0.69±0.08 | **0.80** |
| FrameDiff | 0.86±0.10 | 2.29±1.77 | 0.88±0.02 | 3.54±1.12 | 0.71±0.06 | 0.26 |
| Proteus | 0.94±0.08 | 1.30±1.57 | 0.85±0.13 | * | 0.73±0.07 | 0.46 |
| CarbonNovo w/o plm | 0.58±0.12 | 7.37±3.49 | 0.42±0.08 | * | 0.67±0.04 | 0.58 |
| ProteinWeaver w/o alignment w/o refold | 0.71±0.14 | 4.76±2.91 | 0.62±0.15 | 4.90±1.57 | 0.69±0.06 | 0.66 |
| ProteinWeaver w/o alignment | 0.86±0.11 | 2.59±3.70 | 0.71±0.15 | 4.29±1.19 | 0.66±0.06 | 0.68 |
| ProteinWeaver-sft | 0.86±0.12 | 2.61±2.44 | 0.75±0.14 | 4.16±1.09 | 0.67±0.06 | 0.68 |
| ProteinWeaver w/o bo3 | 0.88±0.13 | 2.47±3.63 | 0.73±0.14 | 4.22±1.03 | **0.65±0.07** | 0.67 |
| ProteinWeaver | 0.92±0.13 | 1.54±0.81 | 0.77±0.12 | 3.78±0.41 | **0.65±0.07** | 0.69 |

**length300**

| | Backbone Quality | | Intf. Quality | | Novelty | Diversity |
|---|---|---|---|---|---|---|
| | scTM ↑ | scRMSD ↓ | Intf. scTM ↑ | Intf. scRMSD ↓ | Max TM ↓ | Max Clust ↑ |
| Native PDB | 0.97±0.10 | 0.82±2.67 | 0.92±0.09 | 1.11±1.45 | – | 0.77 |
| RFDiffusion | 0.89±0.15 | 2.65±3.15 | **0.91±0.11** | **1.65±2.04** | 0.69±0.05 | 0.65 |
| Chroma | 0.83±0.13 | 3.63±3.13 | 0.69±0.18 | 4.56±3.52 | 0.72±0.06 | 0.67 |
| FrameFlow | 0.84±0.15 | 3.56±3.46 | 0.78±0.15 | 4.95±2.28 | 0.72±0.07 | **0.88** |
| FrameDiff | 0.82±0.12 | 3.71±2.69 | 0.85±0.02 | 3.78±0.22 | 0.73±0.06 | 0.21 |
| Proteus | **0.94±0.06** | 1.46±1.08 | 0.89±0.05 | * | 0.78±0.05 | 0.34 |
| CarbonNovo w/o plm | 0.56±0.16 | 9.58±4.69 | 0.52±0.17 | * | 0.74±0.03 | 0.56 |
| ProteinWeaver w/o alignment w/o refold | 0.74±0.12 | 6.10±4.06 | 0.65±0.13 | 5.18±2.31 | 0.69±0.05 | 0.87 |
| ProteinWeaver w/o alignment | 0.86±0.10 | 3.16±2.40 | 0.71±0.16 | 4.48±1.60 | **0.67±0.06** | 0.86 |
| ProteinWeaver-sft | 0.86±0.10 | 3.15±2.20 | 0.74±0.15 | 4.21±1.22 | 0.68±0.06 | 0.85 |
| ProteinWeaver w/o bo3 | 0.88±0.13 | 2.47±3.63 | 0.74±0.07 | 4.29±1.99 | **0.67±0.06** | 0.86 |
| ProteinWeaver | 0.92±0.07 | 2.07±1.32 | 0.77±0.13 | 4.05±1.26 | **0.67±0.06** | 0.86 |

**length500**

| | Backbone Quality | | Intf. Quality | | Novelty | Diversity |
|---|---|---|---|---|---|---|
| | scTM ↑ | scRMSD ↓ | Intf. scTM ↑ | Intf. scRMSD ↓ | Max TM ↓ | Max Clust ↑ |
| Native PDB | 0.97±0.17 | 1.07±5.96 | 0.88±0.17 | 2.23±3.54 | – | 0.8 |
| RFDiffusion | 0.76±0.19 | 5.71±3.53 | 0.80±0.17 | 4.51±3.98 | **0.67±0.04** | 0.89 |
| Chroma | 0.71±0.18 | 8.25±5.73 | 0.52±0.16 | 10.52±4.30 | 0.66±0.07 | **0.99** |
| FrameFlow | 0.66±0.19 | 9.78±5.82 | 0.69±0.17 | 8.20±3.57 | 0.67±0.09 | 0.92 |
| FrameDiff | 0.57±0.23 | 15.61±15.53 | 0.69±0.12 | 6.19±2.00 | 0.68±0.04 | 0.52 |
| Proteus | **0.90±0.13** | 2.76±3.57 | **0.87±0.13** | * | 0.72±0.02 | 0.34 |
| CarbonNovo w/o plm | 0.41±0.09 | 16.02±4.19 | 0.38±0.08 | * | – | 0.76 |
| ProteinWeaver w/o alignment w/o refold | 0.66±0.11 | 9.03±3.99 | 0.63±0.14 | 7.00±2.58 | 0.69±0.06 | 0.81 |
| ProteinWeaver w/o alignment | 0.78±0.14 | 5.77±4.36 | 0.68±0.15 | 5.87±2.93 | 0.67±0.06 | 0.76 |
| ProteinWeaver-sft | 0.72±0.14 | 8.30±3.19 | 0.69±0.09 | 7.15±2.93 | 0.72±0.07 | 0.89 |
| ProteinWeaver w/o bo3 | 0.82±0.10 | 4.39±2.72 | 0.70±0.14 | 5.14±1.77 | **0.67±0.07** | 0.78 |
| ProteinWeaver | 0.86±0.09 | 3.58±2.28 | 0.75±0.15 | **4.28±2.88** | **0.67±0.07** | 0.77 |

**length600**

| | Backbone Quality | | Intf. Quality | | Novelty | Diversity |
|---|---|---|---|---|---|---|
| | scTM ↑ | scRMSD ↓ | Intf. scTM ↑ | Intf. scRMSD ↓ | Max TM ↓ | Max Clust ↑ |
| Native PDB | 0.94±0.07 | 2.03±2.33 | 0.92±0.08 | 2.15±2.01 | – | 0.77 |
| RFDiffusion | 0.66±0.19 | 9.95±5.68 | 0.71±0.16 | 7.30±4.30 | **0.67±0.05** | 0.99 |
| Chroma | 0.62±0.17 | 11.40±5.91 | 0.48±0.12 | 12.97±4.08 | **0.67±0.06** | **1.00** |
| FrameFlow | 0.49±0.14 | 15.82±5.02 | 0.58±0.17 | 9.70±5.19 | 0.69±0.03 | **1.00** |
| FrameDiff | 0.45±0.08 | 16.19±3.35 | 0.53±0.10 | 10.63±2.44 | 0.72±0.03 | **1.00** |
| Proteus | **0.89±0.15** | 3.59±4.18 | **0.89±0.09** | * | 0.68±0.07 | 0.34 |
| CarbonNovo w/o plm | 0.35±0.06 | 19.75±4.40 | 0.33±0.06 | * | – | 0.80 |
| ProteinWeaver w/o alignment w/o refold | 0.57±0.13 | 12.43±4.66 | 0.61±0.14 | 8.28±3.01 | 0.70±0.06 | 0.74 |
| ProteinWeaver w/o alignment | 0.69±0.18 | 8.79±5.74 | 0.69±0.15 | 7.13±3.12 | 0.69±0.06 | 0.76 |
| ProteinWeaver-sft | 0.66±0.16 | 9.71±5.61 | 0.64±0.14 | 8.25±3.44 | 0.69±0.08 | 0.79 |
| ProteinWeaver w/o bo3 | 0.72±0.15 | 5.12±5.19 | 0.70±0.14 | 7.15±2.74 | 0.68±0.07 | 0.75 |
| ProteinWeaver | 0.79±0.12 | 4.95±3.44 | 0.71±0.15 | 6.86±2.89 | 0.68±0.07 | 0.76 |

**length800**

| | Backbone Quality | | Intf. Quality | | Novelty | Diversity |
|---|---|---|---|---|---|---|
| | scTM ↑ | scRMSD ↓ | Intf. scTM ↑ | Intf. scRMSD ↓ | Max TM ↓ | Max Clust ↑ |
| Native PDB | 0.92±0.11 | 2.96±3.47 | 0.93±0.08 | 2.13±2.50 | – | 0.8 |
| RFDiffusion | 0.49±0.12 | 16.80±4.48 | 0.56±0.12 | 12.93±4.01 | **0.66±0.06** | **1.00** |
| Chroma | 0.62±0.14 | 12.75±6.13 | 0.47±0.12 | 14.88±4.37 | 0.68±0.07 | **1.00** |
| FrameFlow | 0.35±0.06 | 23.17±2.55 | 0.58±0.02 | 14.18±5.97 | 0.71±0.02 | **1.00** |
| FrameDiff | 0.38±0.06 | 20.50±3.29 | 0.48±0.10 | 12.20±4.49 | 0.71±0.03 | **1.00** |
| Proteus | 0.67±0.18 | 11.22±6.61 | **0.64±0.18** | * | 0.66±0.04 | 0.56 |
| CarbonNovo w/o plm | 0.25±0.02 | 28.88±4.59 | 0.28±0.07 | * | – | **1.00** |
| ProteinWeaver w/o alignment w/o refold | 0.45±0.08 | 18.56±3.53 | 0.52±0.10 | 13.04±3.39 | 0.70±0.06 | 0.74 |
| ProteinWeaver w/o alignment | 0.54±0.12 | 12.87±4.87 | 0.58±0.09 | 10.56±3.01 | 0.67±0.07 | 0.73 |
| ProteinWeaver-sft | 0.28±0.10 | 31.73±13.14 | 0.54±0.08 | 11.82±2.52 | 0.67±0.04 | 0.95 |
| ProteinWeaver w/o bo3 | 0.63±0.13 | 9.17±5.81 | 0.59±0.14 | 10.28±2.74 | 0.68±0.06 | 0.73 |
| ProteinWeaver | **0.68±0.11** | **8.72±4.60** | 0.60±0.11 | 9.98±2.70 | 0.67±0.06 | 0.73 |

Table 4: Comparison between ProteinWeaver and co-design models evaluated using various lengths ranging from 100 to 800. For each length, we randomly sampled 50 native PDB structures from RCSB as a golden reference for the task. The reported results are the mean±std of repetitive experiments. When the length is 100, the current co-design model generates too few multi-domain proteins and is not statistically significant, so it is not reported.

**length100**

| | Backbone Quality | | Intf. Quality | | Novelty | Diversity |
|---|---|---|---|---|---|---|
| | scTM ↑ | scRMSD ↓ | Intf. scTM ↑ | Intf. scRMSD ↓ | Max TM ↓ | Max Clust ↑ |
| Native PDB | 0.96±0.10 | 0.67±1.61 | 0.82±0.17 | 0.75±0.34 | – | 0.77 |
| Multiflow | **0.96±0.04** | 1.10±0.71 | – | – | 0.71±0.08 | 0.33 |
| CarbonNovo | 0.91±0.14 | 1.16±1.03 | – | – | 0.69±0.09 | 0.71 |
| ProteinWeaver | 0.93±0.06 | **0.99±0.62** | 0.73±0.10 | 3.71±0.37 | **0.61±0.07** | 0.60 |

**length200**

| | Backbone Quality | | Intf. Quality | | Novelty | Diversity |
|---|---|---|---|---|---|---|
| | scTM ↑ | scRMSD ↓ | Intf. scTM ↑ | Intf. scRMSD ↓ | Max TM ↓ | Max Clust ↑ |
| Native PDB | 0.97±0.08 | 0.67±1.37 | 0.85±0.18 | 0.98±1.15 | – | 0.79 |
| Multiflow | **0.95±0.04** | 1.61±1.73 | 0.90±0.03 | * | 0.71±0.07 | 0.42 |
| CarbonNovo | 0.94±0.09 | 1.18±1.47 | **0.97±0.01** | * | 0.71±0.08 | 0.50 |
| ProteinWeaver | 0.92±0.13 | 1.54±0.81 | 0.77±0.12 | 3.78±0.41 | **0.65±0.07** | 0.69 |

**length300**

| | Backbone Quality | | Intf. Quality | | Novelty | Diversity |
|---|---|---|---|---|---|---|
| | scTM ↑ | scRMSD ↓ | Intf. scTM ↑ | Intf. scRMSD ↓ | Max TM ↓ | Max Clust ↑ |
| Native PDB | 0.97±0.10 | 0.82±2.67 | 0.92±0.09 | 1.11±1.45 | – | 0.77 |
| Multiflow | **0.96±0.06** | 2.14±3.24 | 0.91±0.04 | * | 0.71±0.06 | 0.58 |
| CarbonNovo | 0.95±0.08 | **1.33±1.59** | **0.93±0.11** | * | 0.74±0.05 | 0.31 |
| ProteinWeaver | 0.92±0.07 | 2.07±1.32 | 0.77±0.13 | 4.05±1.26 | **0.67±0.06** | **0.86** |

**length500**

| | Backbone Quality | | Intf. Quality | | Novelty | Diversity |
|---|---|---|---|---|---|---|
| | scTM ↑ | scRMSD ↓ | Intf. scTM ↑ | Intf. scRMSD ↓ | Max TM ↓ | Max Clust ↑ |
| Native PDB | 0.97±0.17 | 1.07±5.96 | 0.88±0.17 | 2.23±3.54 | – | 0.8 |
| Multiflow | 0.83±0.10 | 8.48±5.32 | **0.84±0.07** | * | 0.68±0.06 | 0.67 |
| CarbonNovo | 0.85±0.15 | 4.07±4.14 | 0.83±0.17 | * | 0.68±0.05 | 0.67 |
| ProteinWeaver | **0.86±0.09** | 3.58±2.28 | 0.75±0.15 | 4.28±2.88 | **0.67±0.07** | **0.77** |

**length600**

| | Backbone Quality | | Intf. Quality | | Novelty | Diversity |
|---|---|---|---|---|---|---|
| | scTM ↑ | scRMSD ↓ | Intf. scTM ↑ | Intf. scRMSD ↓ | Max TM ↓ | Max Clust ↑ |
| Native PDB | 0.94±0.07 | 2.03±2.33 | 0.92±0.08 | 2.15±2.01 | – | 0.77 |
| Multiflow | 0.61±0.13 | 12.41±4.74 | 0.49±0.11 | * | 0.71±0.07 | 0.62 |
| CarbonNovo | **0.87±0.09** | **4.20±4.09** | **0.81±0.09** | * | 0.70±0.06 | **0.93** |
| ProteinWeaver | 0.79±0.12 | 4.95±3.44 | 0.71±0.15 | 6.86±2.89 | **0.68±0.07** | 0.76 |

**length800**

| | Backbone Quality | | Intf. Quality | | Novelty | Diversity |
|---|---|---|---|---|---|---|
| | scTM ↑ | scRMSD ↓ | Intf. scTM ↑ | Intf. scRMSD ↓ | Max TM ↓ | Max Clust ↑ |
| Native PDB | 0.92±0.11 | 2.96±3.47 | 0.93±0.08 | 2.13±2.50 | – | 0.8 |
| Multiflow | 0.37±0.07 | 25.86±3.18 | 0.27±0.06 | * | – | 0.54 |
| CarbonNovo | 0.52±0.13 | 16.53±5.39 | 0.57±0.17 | * | **0.67±0.03** | **1.00** |
| ProteinWeaver | **0.68±0.11** | **8.72±4.60** | 0.60±0.11 | 9.98±2.70 | 0.67±0.06 | 0.73 |

Table 5: The scTM calculation involves generating eight sequences for each designed backbone using proteinMPNN, followed by structure prediction with ESMFold for each sequence. The sequence with the highest scTM score is then selected for further analysis. As both the number of sampled sequences and the length of the designed proteins increase, the computational demands rise significantly, as illustrated in the table below.

| length | seq_per_sample (unit: second) | | | |
|---|---|---|---|---|
| | 1 | 2 | 4 | 8 |
| 50 | 9.9 | 12.3 | 16.3 | 23 |
| 100 | 10.9 | 12 | 17.2 | 25 |
| 200 | 16 | 21.3 | 32 | 52 |
| 300 | 30 | 41 | 66 | 113 |
| 400 | 51 | 76 | 132 | 216 |
| 500 | 89 | 130 | 211 | 377 |

Table 6: Evaluation of Preference Alignment Methods. We evaluated the performance of SPPO with different preference alignment methods: SFT (Supervised Fine-Tuning) and SFT + DPO (Direct Preference Optimization). To ensure the robustness of our implementation, we tested DPO with various beta parameters. The beta parameter controls the degree of difference between the fine-tuned model and the reference model. A larger beta value makes it less likely for the model to deviate from the reference model during the fine-tuning process.

| | length100 | | | | length200 | | | |
|---|---|---|---|---|---|---|---|---|
| | scTM ↑ | scRMSD ↓ | Max TM ↓ | Max Clust ↑ | scTM ↑ | scRMSD ↓ | Max TM ↓ | Max Clust ↑ |
| SPPO | **0.91±0.14** | 1.27±1.84 | **0.61±0.07** | 0.59 | **0.88±0.13** | 2.47±3.63 | **0.65±0.07** | 0.67 |
| SFT | **0.91±0.12** | **1.20±1.79** | 0.62±0.06 | 0.61 | 0.86±0.12 | 2.61±2.44 | 0.67±0.06 | 0.68 |
| SFT+DPO (beta=10) | 0.90±0.11 | 1.45±1.96 | 0.62±0.04 | **0.62** | 0.87±0.13 | 2.57±2.41 | 0.67±0.06 | 0.68 |
| SFT+DPO (beta=1.0) | 0.88±0.14 | 1.89±2.18 | 0.62±0.05 | **0.62** | 0.83±0.13 | 2.93±2.41 | 0.68±0.07 | 0.68 |
| SFT+DPO (beta=0.1) | 0.88±0.14 | 1.89±2.18 | 0.63±0.05 | **0.62** | 0.78±0.14 | 3.74±2.83 | 0.67±0.05 | **0.72** |

| | length300 | | | | length500 | | | |
|---|---|---|---|---|---|---|---|---|
| | scTM ↑ | scRMSD ↓ | Max TM ↓ | Max Clust ↑ | scTM ↑ | scRMSD ↓ | Max TM ↓ | Max Clust ↑ |
| SPPO | **0.88±0.13** | **2.47±3.63** | **0.67±0.06** | 0.86 | **0.82±0.10** | 4.39±2.72 | **0.67±0.07** | 0.78 |
| SFT | 0.86±0.10 | 3.15±2.20 | 0.68±0.06 | 0.85 | 0.72±0.14 | 8.30±3.19 | 0.72±0.07 | 0.89 |
| SFT+DPO (beta=10) | 0.85±0.09 | 3.24±2.37 | 0.68±0.06 | 0.85 | 0.72±0.13 | 8.26±3.05 | 0.72±0.07 | 0.89 |
| SFT+DPO (beta=1.0) | 0.86±0.10 | 3.02±2.05 | 0.68±0.06 | 0.85 | 0.77±0.13 | 5.79±3.66 | 0.72±0.06 | 0.61 |
| SFT+DPO (beta=0.1) | 0.80±0.12 | 4.10±2.91 | 0.70±0.03 | **0.87** | 0.48±0.09 | 12.99±3.04 | 0.78±0.02 | **0.97** |

| | length600 | | | | length800 | | | |
|---|---|---|---|---|---|---|---|---|
| | scTM ↑ | scRMSD ↓ | Max TM ↓ | Max Clust ↑ | scTM ↑ | scRMSD ↓ | Max TM ↓ | Max Clust ↑ |
| SPPO | **0.72±0.15** | **7.15±2.74** | **0.68±0.07** | 0.75 | **0.63±0.13** | **9.17±5.81** | 0.68±0.06 | 0.73 |
| SFT | 0.66±0.16 | 9.71±5.61 | 0.69±0.05 | 0.79 | 0.28±0.16 | 31.73±13.14 | **0.67±0.04** | 0.95 |
| SFT+DPO (beta=10) | 0.68±0.14 | 9.40±4.91 | 0.69±0.05 | 0.79 | 0.34±0.22 | 27.11±11.90 | 0.68±0.036 | 0.93 |
| SFT+DPO (beta=1.0) | 0.67±0.15 | 9.24±4.72 | 0.71±0.06 | 0.68 | 0.55±0.10 | 13.89±4.54 | 0.70±0.03 | 0.85 |
| SFT+DPO (beta=0.1) | 0.40±0.07 | 20.24±12.88 | 0.76±0.03 | **1.00** | 0.27±0.10 | 33.50±15.72 | – | **1.00** |

Table 7: We explored the impact of varying reward assignments by adjusting preferences from scTM to interface scTM, as well as combining both metrics. For the combined approach, we selected the sample that exhibited the best performance in both scTM and interface scTM. We believe that scTM serves as a general quality metric for backbone structures, while interface scTM focuses on optimizing inter-domain interactions.

| | length100 | | | | | length200 | | | | |
|---|---|---|---|---|---|---|---|---|---|---|
| | scTM ↑ | scRMSD ↓ | Intf. scTM ↑ | Max TM ↓ | Max Clust ↑ | scTM ↑ | scRMSD ↓ | Intf. scTM ↑ | Max TM ↓ | Max Clust ↑ |
| ProteinWeaver (scTM) w/o bo3 | **0.91±0.14** | 1.27±1.84 | 0.75±0.14 | **0.61±0.07** | 0.59 | **0.88±0.13** | 2.47±3.63 | 0.73±0.14 | 0.65±0.07 | 0.67 |
| ProteinWeaver (interface scTM) w/o bo3 | 0.86±0.11 | 1.73±1.74 | 0.60±0.12 | 0.62±0.08 | **0.61** | 0.84±0.12 | 2.69±2.56 | 0.71±0.15 | 0.66±0.05 | **0.69** |
| ProteinWeaver (scTM + interface scTM) w/o bo3 | **0.91±0.08** | **1.12±0.81** | **0.77±0.14** | **0.61±0.06** | 0.60 | **0.88±0.12** | **2.14±2.07** | **0.76±0.14** | **0.65±0.06** | 0.67 |

| | length300 | | | | | length500 | | | | |
|---|---|---|---|---|---|---|---|---|---|---|
| | scTM ↑ | scRMSD ↓ | Intf. scTM ↑ | Max TM ↓ | Max Clust ↑ | scTM ↑ | scRMSD ↓ | Intf. scTM ↑ | Max TM ↓ | Max Clust ↑ |
| ProteinWeaver (scTM) w/o bo3 | **0.88±0.13** | **2.47±3.63** | **0.74±0.07** | 0.67±0.06 | 0.86 | 0.82±0.10 | 4.39±2.72 | 0.70±0.14 | **0.67±0.07** | 0.78 |
| ProteinWeaver (interface scTM) w/o bo3 | 0.82±0.12 | 3.79±2.89 | 0.66±0.17 | **0.66±0.06** | 0.83 | 0.79±0.14 | 5.23±3.89 | 0.67±0.15 | 0.69±0.08 | **0.79** |
| ProteinWeaver (scTM + interface scTM) w/o bo3 | 0.86±0.10 | 2.66±2.21 | **0.74±0.14** | **0.66±0.06** | **0.88** | **0.84±0.12** | **4.21±3.33** | **0.72±0.14** | 0.68±0.06 | 0.78 |

| | length600 | | | | | length800 | | | | |
|---|---|---|---|---|---|---|---|---|---|---|
| | scTM ↑ | scRMSD ↓ | Intf. scTM ↑ | Max TM ↓ | Max Clust ↑ | scTM ↑ | scRMSD ↓ | Intf. scTM ↑ | Max TM ↓ | Max Clust ↑ |
| ProteinWeaver (scTM) w/o bo3 | **0.72±0.15** | **7.15±2.74** | **0.70±0.14** | 0.68±0.07 | 0.75 | **0.63±0.13** | **9.17±5.81** | **0.59±0.14** | **0.68±0.06** | 0.73 |
| ProteinWeaver (interface scTM) w/o bo3 | 0.69±0.14 | 8.79±5.14 | 0.61±0.17 | **0.67±0.07** | **0.78** | 0.54±0.11 | 14.36±4.35 | 0.52±0.13 | 0.70±0.09 | **0.88** |
| ProteinWeaver (scTM + interface scTM) w/o bo3 | **0.72±0.13** | 7.28±4.72 | 0.68±0.15 | **0.67±0.06** | 0.76 | 0.62±0.09 | 9.34±4.88 | 0.57±0.15 | **0.68±0.08** | 0.75 |

Table 8: We have also experimented with adding triangular attention to ProteinWeaver and evaluated its performance. We replaced the edge transition from MLP to triangular attention, and the performance results are shown in below.

| Model | length100 | | | | | length200 | | | | |
|---|---|---|---|---|---|---|---|---|---|---|
| | scTM | scRMSD | Interface scTM | diversity | novelty | scTM | scRMSD | Interface scTM | diversity | novelty |
| ProteinWeaver(MLP) | 0.90±0.14 | **1.26±1.97** | 0.73±0.07 | **0.60** | **0.61±0.07** | 0.86±0.11 | 2.59±3.70 | 0.71±0.15 | **0.68** | 0.66±0.06 |
| ProteinWeaver(Triangular attention) | **0.92±0.11** | 1.66±1.70 | **0.76±0.12** | 0.58 | 0.62±0.10 | **0.88±0.10** | **1.99±1.47** | **0.77±0.12** | 0.67 | **0.67±0.06** |

| Model | length300 | | | | | length500 | | | | |
|---|---|---|---|---|---|---|---|---|---|---|
| | scTM | scRMSD | Interface scTM | diversity | novelty | scTM | scRMSD | Interface scTM | diversity | novelty |
| ProteinWeaver(MLP) | 0.86±0.10 | 3.16±2.40 | 0.71±0.16 | **0.86** | **0.67±0.06** | 0.78±0.14 | 5.77±4.36 | **0.68±0.15** | **0.76** | 0.67±0.06 |
| ProteinWeaver(Triangular attention) | **0.88±0.09** | **2.64±1.96** | **0.73±0.13** | 0.85 | 0.70±0.07 | **0.80±0.12** | **5.35±3.96** | 0.68±0.19 | 0.72 | 0.68±0.06 |

| Model | length600 | | | | | length800 | | | | |
|---|---|---|---|---|---|---|---|---|---|---|
| | scTM | scRMSD | Interface scTM | diversity | novelty | scTM | scRMSD | Interface scTM | diversity | novelty |
| ProteinWeaver(MLP) | **0.69±0.18** | 8.79±5.74 | **0.69±0.15** | 0.76 | 0.69±0.06 | 0.54±0.12 | 12.87±4.87 | **0.58±0.09** | 0.73 | 0.67±0.07 |
| ProteinWeaver(Triangular attention) | 0.69±0.19 | **8.52±5.56** | 0.66±0.12 | 0.76 | **0.68±0.08** | **0.56±0.11** | 12.18±4.43 | 0.57±0.09 | 0.73 | 0.67±0.07 |

