# OpenReview forum: "ProteinWeaver: A Divide-and-Assembly Approach for Protein Backbone Design"
_ICLR.cc/2025/Conference — ICLR 2025 Conference Desk Rejected Submission_

### Official Review · Reviewer_tTLY · 2024-11-01

**Soundness:** 2
**Presentation:** 3
**Contribution:** 2
**Rating:** 3
**Confidence:** 5

**Summary:**

This paper proposes a domain assembly framework, coupled with preference alignment, to flexibly assemble multiple protein domains, achieving better generation performance on long proteins compared to existing protein backbone design models.

**Strengths:**

The paper is generally clear and easy to understand. The concept of flexibly folding multiple domains is an interesting application of recent protein diffusion models and could be very useful for designing more complex functional proteins.

**Weaknesses:**

Besides the limitations mentioned in Section 5, one major concern is the fairness of comparisons with other protein backbone models. Specifically, while other protein design models such as RFDiffusion and Chroma generate proteins from scratch (unconditional generation), ProteinWeaver performs a conditional generation task with substantial “local” information (individual domain structures) provided. This combines with observation that individual structures do not shift substantially (with mean alterations less than 2A as shown in Figure 10), suggesting that ProteinWeaver tackles an easier task compared to other protein backbone models and thus it is not a fair comparison with other models such as RFDiffusion and Chroma. In addition, details on the generation of samples that are used for comparison with other backbone design models are lacking, for example, what existing backbone design models are used for generation, how many domains are generated by each model, how the generation is decomposed (percentage of each decomposition) and so on. These details are also crucial for understanding the novelty argument in Section 3.3; in particular, to what extent is the improvement in novelty a result of higher novelty in the underlying domain pool, or a result of the assembly process?

Considering the limited technical novelty in model architecture (FrameDiff with minor adaptations), it would be helpful if the authors could provide more ablation analysis on the preference alignment component. For example, what is the effect of different reward assignments? How would the performance change if the preference considers more than just scTM, but also interface scTM and other metrics? How do various factors on the alignment dataset construction (for example, size, diversity in domain selection) affect model performance?

**Questions:**

- In Appendix A.5, the description on the computation of scTM and scRMSD differs from the diagram in Figure 8. Could the authors clarify further on this? Is the ProteinMPNN-ESMFold pipeline used for refolding the backbone or the interface structure?
- In Figure 3, why does assembly using PDB domain perform better than assembly using CATH domain? For assembly using PDB domain, does the model perform assembly using domains from the same PDB structure, leading to better performance over assembly using CATH domains? Moreover, for assembly with synthesized domains, does the model perform assembly using domains from the same synthesized structure? Why is the performance of assembly using RFDiffusion-generated domains better than that of PDB and CATH domains? How many structures are generated for each method? This same set of questions also applies for Figure 6.
- In Figure 1F, what is the scale on each axis?
- In line 257-265, what are the criteria used for deduplication and domain selection?
- In Section 3.5, it would be useful if the authors could provide further details on the refolding of domains. Specifically, to what extent is the structure altered in the refolding process? What is the self-consistency performance for initial domains (directly split from PDB) and refolded domains? Does the domain in the final assembly resemble more closely to the initial domain or the refolded domain?
- Could the authors provide further details on the best-of-3 filter? In Figure 7, the performance of ProteinWeaver without alignment and bo3 is close to the performance of ProteinWeaver. Interestingly, the performance of ProteinWeaver without alignment and bo3 seems to outperform the performance of ProteinWeaver without alignment (but with bo3 filter). Do the authors have any intuition on these observations? Moreover, the margin of benefits that alignment provides seems to be smaller than illustrated in Figure 6. It would be helpful if the authors could also show the performance of ProteinWeaver in Figure 6.
- For comparisons with backbone design models, it is worth noting that RFDiffusion is trained with a crop size of 384 residues, which is smaller than that used for training ProteinWeaver. This could give ProteinWeaver an edge in generating longer proteins since the training dataset consists of more long proteins. In addition, how does the model perform compared to Proteus (https://www.biorxiv.org/content/10.1101/2024.02.10.579791v1), which exhibits better generation performance on long proteins than RFDiffusion?
- For pretraining, does the model start from scratch or pretrained FrameDiff weights? Also, it would be helpful if the authors could provide further details on the training and sampling time: How long does it take to pretrain the model and how many epochs does it run? How long does it take to align the model?
- Figure 9: How does ProteinWeaver perform here?
- How does ProteinWeaver perform as the number of domains increases?

---

> ### Author Response · Authors · 2024-11-22
> **Rebuttal by Authors**
>
> We greatly appreciate your constructive feedback regarding several aspects of our manuscript, including contributions of this study, implementation details, insufficient ablation studies. To address your concerns, we have implemented mainly three improvements:
>
> -We explained and emphasized the contributions of our study.
> -We have provided more implementation details in the revised manuscript.
> -We have provided extended ablation studies for alignment.
>
> We believe these revisions have substantially strengthened the manuscript and welcome your assessment of these changes.
>
> **[Q1]** Besides the limitations mentioned in Section 5, one major concern is the fairness of comparisons with other protein backbone models. Specifically, while other protein design models such as RFDiffusion and Chroma generate proteins from scratch (unconditional generation), ProteinWeaver performs a conditional generation task with substantial “local” information (individual domain structures) provided. This combines with observation that individual structures do not shift substantially (with mean alterations less than 2A as shown in Figure 10), suggesting that ProteinWeaver tackles an easier task compared to other protein backbone models and thus it is not a fair comparison with other models such as RFDiffusion and Chroma.
>
> **[A1]** We appreciate the reviewer’s thoughtful comments. However, **we respectfully disagree with the characterization of our contribution as unfair** and would like to clarify several key points:
>
> 1. **Scientific Innovation**
>  There seems to be a misunderstanding regarding the contributions of our study. **Domain assembly is a well-established task in protein design that has been extensively researched using traditional computational biology methods, such as Rosetta. Our study provides the first paradigm introducing a general domain-assembly approach based on deep-learning methods.**
>
> 2. **Methodology**
>  **Our work builds upon existing foundations in protein design while introducing novel methodological advances. Similar to successful hierarchical generation and divide-and-conquer strategies used in many computer vision studies [1-2], we apply these principles to protein design. ProteinWeaver represents the first deep learning approach for domain assembly-based protein design, establishing a new paradigm that encompasses both backbone design and functional design.**
>
> 3. **Fair Performance Comparison**
> While the reviewer suggests that using domain information makes our task "easier," this perspective overlooks several important aspects. **Maintaining structural integrity while assembling domains is a non-trivial challenge, and our method effectively addresses both local and global structural constraints. Additionally, our study demonstrates significant performance enhancements specifically for long-chain backbone design, highlighting the effectiveness of our approach.**
>
> 4. **Scope and Applications**
> We acknowledge that we may have overemphasized the backbone design aspect in our original manuscript.  And we will tune down in the revised manuscript. However, **the contributions of our work extend well beyond this focus. We have introduced a novel framework for protein design based on domain assembly, which not only demonstrates superior performance in backbone design tasks but also includes functional design capabilities, particularly for dimeric binder design.**
>
> [1] Ho J, Saharia C, Chan W, et al. Cascaded diffusion models for high fidelity image generation[J]. Journal of Machine Learning Research, 2022, 23(47): 1-33.
>
> [2] Bar-Tal O, Yariv L, Lipman Y, et al. Multidiffusion: Fusing diffusion paths for controlled image generation[J]. 2023.
>
> **[Q2]** In addition, details on the generation of samples that are used for comparison with other backbone design models are lacking, for example, what existing backbone design models are used for generation, how many domains are generated by each model, how the generation is decomposed (percentage of each decomposition) and so on. These details are also crucial for understanding the novelty argument in Section 3.3; in particular, to what extent is the improvement in novelty a result of higher novelty in the underlying domain pool, or a result of the assembly process?
>
> **[A2]**We thank the reviewer for the comment. We fulliy agree more details will add the transparency of our study. In the revised manuscript, we add more details in the appendix.

---

> ### Author Response · Authors · 2024-11-22
> **Rebuttal by Authors**
>
> **[Q3]** Considering the limited technical novelty in model architecture (FrameDiff with minor adaptations), it would be helpful if the authors could provide more ablation analysis on the preference alignment component. For example, what is the effect of different reward assignments? How would the performance change if the preference considers more than just scTM, but also interface scTM and other metrics? How do various factors on the alignment dataset construction (for example, size, diversity in domain selection) affect model performance?
>
> **[A3]**We thank the reviewer for raising these important points. We agree that comprehensive ablation studies are crucial for assessing the technical contributions of our study, and we have conducted extensive analyses on the preference alignment component.
>
>
> 1. **Evaluation of Preference Alignment Methods**
> **We evaluated the performance of SPPO with different preference alignment methods: SFT (Supervised Fine-Tuning) and SFT + DPO (Direct Preference Optimization).** To ensure the robustness of our implementation, we tested DPO with various beta parameters. The beta parameter controls the degree of difference between the fine-tuned model and the reference model. A larger beta value makes it less likely for the model to deviate from the reference model during the fine-tuning process.
>
> The results, shown in the updated Table 6 & 7 in the revised manuscript, demonstrate that **SPPO-based alignment consistently outperforms both SFT and SFT + DPO in terms of design quality and novelty across different protein lengths. Notably, the performance advantage of SPPO increases significantly with longer protein sequences, indicating its effectiveness for long-chain protein design.** While SFT + DPO with a beta of 0.1 enhances the diversity of designed structures, it leads to a marked decline in design quality as measured by scTM and scRMSD. This finding highlights an **inverse relationship between structural quality and diversity: as structural quality decreases, diversity tends to increase.**
>
> --Mean result of short length:
> | model              | length100,200,300 |               |           |               |
> |--------------------|:-----------------:|:-------------:|:---------:|:-------------:|
> |                    | scTM              | scRMSD        | diversity | novelty       |
> | SPPO               | **0.91±0.14**     | 1.27±1.84     | 0.59      | **0.61±0.07**     |
> | SFT                | **0.91±0.12**     | **1.20±1.79** | 0.61      | 0.62±0.06     |
> | SFT+DPO (beta=10)  | 0.90±0.11         | 1.45±1.96     | **0.62**  | 0.62±0.04     |
> | SFT+DPO (beta=1)   | 0.88±0.14         | 1.89±2.18     | **0.62**  | 0.62±0.05     |
> | SFT+DPO (beta=0.1) | 0.88±0.14         | 1.89±2.18     | **0.62**  | 0.63±0.05 |
>
> --Mean result of long length:
>
> | model              | length 500, 600, 800 |         |           |          |
> |--------------------|:--------------------:|:-------:|:---------:|:--------:|
> |                    | scTM                 |  scRMSD | diversity | novelty  |
> | SPPO               | **0.72**             | **6.9** | 0.75      | **0.68**     |
> | SFT                | 0.55                 | 16.58   | 0.88      | 0.69     |
> | SFT+DPO (beta=10)  | 0.58                 | 14.92   | 0.87      | 0.7      |
> | SFT+DPO (beta=1)   | 0.66                 | 9.64    | 0.71      | 0.71     |
> | SFT+DPO (beta=0.1) | 0.38                 | 22.24   | **0.99**  | 0.75 |

---

> ### Author Response · Authors · 2024-11-22
> **Part 2 for Q3**
>
> 2. **Different reward assignments**
>
> **We explored the impact of varying reward assignments by adjusting preferences from scTM to interface scTM, as well as combining both metrics.** For the combined approach, we selected the sample that exhibited the best performance in both scTM and interface scTM. We believe that scTM serves as a general quality metric for backbone structures, while interface scTM focuses on optimizing inter-domain interactions. The results, presented in the accompanying tables (Table 7 in the revised manuscript), indicate that for various protein lengths, **structures optimized using scTM yield superior quality compared to those optimized with interface scTM.** This suggests that global quality alignment via scTM is more effective than local interface alignment.
>
> Additionally, **using a combined reward assignment of scTM and interface scTM resulted in a slight improvement in interface scTM scores compared to using scTM**, with only marginal changes in overall scTM scores. This suggests that **scTM is sufficient for optimizing structural quality.**
>
> | model                                         | short length (mean result of 100, 200, 300) |          |                |           |          |
> |-----------------------------------------------|:-------------------------------------------:|:--------:|:--------------:|:---------:|:--------:|
> |                                               | scTM                                        |  scRMSD  | interface scTM | diversity | novelty  |
> | ProteinWeaver (scTM) w/o bo3                  |                   **0.89**                  |   2.07   |      0.74      |    0.71   | **0.64** |
> | ProteinWeaver (interface scTM) w/o bo3        |                     0.84                    |   2.74   |      0.66      |    0.71   |   0.65   |
> | ProteinWeaver (scTM + interface scTM) w/o bo3 |                     0.88                    | **1.97** |    **0.76**    |  **0.72** | **0.64** |
>
>
> | model                                         | long length (mean result of 500, 600, 800) |         |                |           |          |
> |-----------------------------------------------|:------------------------------------------:|:-------:|:--------------:|:---------:|:--------:|
> |                                               | scTM                                       |  scRMSD | interface scTM | diversity | novelty  |
> | ProteinWeaver (scTM) w/o bo3                  |                    0.72                    | **6.9** |    **0.66**    |    0.75   | **0.68** |
> | ProteinWeaver (interface scTM) w/o bo3        |                    0.67                    |   9.46  |       0.6      |  **0.82** |   0.69   |
> | ProteinWeaver (scTM + interface scTM) w/o bo3 |                  **0.73**                  |   6.94  |    **0.66**    |    0.76   | **0.68** |

---

> ### Author Response · Authors · 2024-11-22
> **Rebuttal by Authors**
>
> **[Q4]** In Appendix A.5, the description on the computation of scTM and scRMSD differs from the diagram in Figure 8. Could the authors clarify further on this? Is the ProteinMPNN-ESMFold pipeline used for refolding the backbone or the interface structure?
>
> **[A4]** Thank you for pointing out this apparent discrepancy. The diagram in Figure 8 specifically shows the computation of scTM and scRMSD for interface regions, while Appendix A.5 describes the general computation process. To avoid confusion, we will revise Appendix A.5 to explicitly differentiate between interface-specific metrics (as shown in Figure 8) and Full structure evaluation metrics.
>
> **[Q5]** In Figure 3, why does assembly using PDB domain perform better than assembly using CATH domain? For assembly using PDB domain, does the model perform assembly using domains from the same PDB structure, leading to better performance over assembly using CATH domains? Moreover, for assembly with synthesized domains, does the model perform assembly using domains from the same synthesized structure? Why is the performance of assembly using RFDiffusion-generated domains better than that of PDB and CATH domains? How many structures are generated for each method? This same set of questions also applies for Figure 6.
>
> **[A5]** To answer this question, **we provide an analysis of structural variation Impact**.
>
> **we provide an analysis to illustrate the impact of structural variations in domains on assembly quality**. We clustered natural domains in the CATH dataset based on their topological differences, resulting in approximately 530 domain structures that represent a **natural distribution of protein domains**. This dataset allows us to evaluate the effects of structural variations effectively.
>
> We performed **pairwise assemblies of all these domains** and assessed the quality of the designed structures using the scTM score. Our analysis included a **comparison of secondary structure ratios at the assembly interface**, which we believe directly affects domain assembly.
>
> The results presented in Figure 11 in Appendix confirm our expectations: both the type and ratio of secondary structures significantly influence the designability of assembled structures. Stable secondary structures, such as alpha helices and beta strands, are optimal for domain assembly. We found that increasing the ratio of either helices or strands at the interface enhances designability. For instance, as the helix ratio increases from 0% to 90%, the designability measure (scTM) rises from 0.5 to 0.7. Similarly, for beta strands, the scTM increases from 0.56 to 0.63.
>
> **Helices are easier to assemble than strands, with improvements in designability becoming more pronounced as the ratio of helices increases. In contrast, loops—being unstable—significantly impair assembly quality, with scTM dropping from 0.87 to 0.46. This observation provides valuable insights into the principles of native domain assembly.**
>
> **Based on this observation, we speculate the structural variations may induce different assembly performance on PDB, CATH and RFdiffusion. To specifically answer the reviewer's question, we are working on statistically analyzing the structural variances for these datasets. We expect to get the result in the next few days.**
>
> **[Q6]** In Figure 1F, what is the scale on each axis?
>
> **[A6]** In Figure 1F, we used the indicators scTM, scRMSD, Interface scTM to evaluate quality, Max Clust to evaluate diversity, and Max TM to evaluate novelty. Except for scRMSD, all evaluation indicators are values between 0 and 1. When drawing scRMSD, we use PDB as the gold standard and use the average scRMSD of the PDB structures divided by the average scRMSD of the backbone model generated structures as the final value for drawing.
>
> **[Q7]** In line 257-265, what are the criteria used for deduplication and domain selection?
>
> **[A7]** When constructing SPPO data, we use the scTM of the structure as a criterion to select winner data and loser data.
>
> **[Q8]** In Section 3.5, it would be useful if the authors could provide further details on the refolding of domains. Specifically, to what extent is the structure altered in the refolding process? What is the self-consistency performance for initial domains (directly split from PDB) and refolded domains? Does the domain in the final assembly resemble more closely to the initial domain or the refolded domain?
>
> **[A8]** We found that the single domain structure obtained through refold has mainly changed in the interface area compared to its structure in the PDB. We calculated the TMscore of the structures before and after our refold and found that the average TMscore before and after refold was 0.85, and the standard deviation was 0.15. In addition, we found that the assembled structure is closer to the original structure, which we guess is related to the fact that we use the real structure in the PDB as the training target.

---

> ### Author Response · Authors · 2024-11-22
> **Rebuttal by Authors**
>
> **[Q9]** Could the authors provide further details on the best-of-3 filter? In Figure 7, the performance of ProteinWeaver without alignment and bo3 is close to the performance of ProteinWeaver. Interestingly, the performance of ProteinWeaver without alignment and bo3 seems to outperform the performance of ProteinWeaver without alignment (but with bo3 filter). Do the authors have any intuition on these observations? Moreover, the margin of benefits that alignment provides seems to be smaller than illustrated in Figure 6. It would be helpful if the authors could also show the performance of ProteinWeaver in Figure 6.
>
> **[A9]** We thanks for the question. There is a misunderstanding for the results. In figure 6, we applied ProteinWeaver for different datasets including PDB, CATH. In Figure 7, we only used bo3 in the last method named "ProteinWeaver". For the other methods, we didn't use bo3.

---

> ### Author Response · Authors · 2024-11-22
> **Rebuttal of Authors**
>
> **[Q10]** For comparisons with backbone design models, it is worth noting that RFDiffusion is trained with a crop size of 384 residues, which is smaller than that used for training ProteinWeaver. This could give ProteinWeaver an edge in generating longer proteins since the training dataset consists of more long proteins. In addition, how does the model perform compared to Proteus (https://www.biorxiv.org/content/10.1101/2024.02.10.579791v1), which exhibits better generation performance on long proteins than RFDiffusion?
>
> we have **expanded our analysis** to include additional state-of-the-art methods in both the backbone design (Proteus) and sequence-structure codesign (CarbonNovo and MultiFlow) categories. **It is important to note that sequence-structure codesign methods leverage both sequence and structure datasets for model training, which makes direct comparisons with our backbone design methods—focused solely on backbone information—unfair. Nonetheless, we have included comparisons with these methods.**
>
> We provide a brief introduction of these methods. A key limitation shared by all of these approaches is that they fail to account for the hierarchical, multi-domain nature of long-chain proteins.
>
> 1. **Proteus:** A state-of-the-art backbone design method that employs AlphaFold2's graph-based triangle approach and multi-track interaction networks. It outperforms RFdiffusion in long-chain monomer (>400 residues) generation and exceeds Chroma in complex structure generation.
>
> 2. **MultiFlow:** A sequence-structure codesign method using discrete flow-based generative models. It demonstrates superior designability compared to RFdiffusion, partly due to its **knowledge distillation from ProteinMPNN.**
>
> 3. **CarbonNovo:** An end-to-end sequence-structure codesign method using a unified energy-based model. It pioneered the **integration of protein language models (pLM)** for both structure and sequence generation, showing substantial improvement in designability (from 51.75% to 81.38%).
>
> More detailed comparative results of this analysis are presented in Table 3 & 4 of the revised manuscript.
>
> 1. For short-chain protein backbones (100-300 residues), all methods demonstrate the ability to generate high-quality backbones, consistent with our original manuscript. RFdiffusion shows slightly superior design quality, while ProteinWeaver achieves the highest novelty scores, both with and without alignment.
>
> Table showing the mean results for 100, 200, 300
> |                 | model                                              | scTM | scRMSD | Interface scTM | diversity | novelty |
> |-----------------|----------------------------------------------------|------|--------|----------------|-----------|---------|
> | Backbone design | Native PDBs                                        | 0.97 |  0.72  |      0.86      |    0.77   |    --   |
> |                 | RFdiffusion                                        | **0.98** |  **0.71**  |       0.9      |    0.54   |   0.72  |
> |                 | Proteus                                            | 0.94 |   1.2  |      0.87      |    $\color{red}{\text{0.43}}$   |   0.75  |
> |                 | CarbonNovo w/o plm                                 | 0.59 |  7.07  |      0.47      |    0.69   |   0.69  |
> |                 | ProteinWeaver w/o alignment                        | 0.87 |  2.34  |      0.72      |    0.71   |   **0.64**  |
> |                 | ProteinWeaver w/o alignment + triangular attention | 0.89 |   2.1  |      0.75      |    0.70    |   0.66  |
> |                 | ProteinWeaver w/ alignment                         | 0.92 |  1.53  |      0.75      |    **0.72**   |   **0.64**  |
> | Co-design       | Multiflow*                                         | 0.96 |  1.62  |      0.91      |    0.44   |   0.71  |
> |                 | CarbonNovo*                                        | 0.93 |  1.22  |      **0.95**      |    0.51   |   0.71  |

---

> > ### Author Response · Authors · 2024-11-22
> > **Extended response for Q10**
> >
> > 2. **In the case of long-chain protein backbones (500-800 residues), ProteinWeaver demonstrates significantly higher scTM scores compared to RFdiffusion, MultiFlow, and CarbonNovo (without protein language model). Additionally, ProteinWeaver outperforms CarbonNovo (with protein language model) in scTM for 800 residue backbone design.**
> >
> > 3. While **Proteus achieves the best design quality metrics (scTM and scRMSD), it shows significantly reduced structural diversity compared to other methods** (highlighted in $\color{red}{\text{red}}$). Case analysis reveals that Proteus generates structures limited to three categories, primarily helical tandem repeats, confirming our diversity metric findings.
> >
> > 4. An inverse relationship between scTM with diversity is observed. For 600 and 800 length, RFdiffusion's scTM decreases and diversity increases. The diversity score even exceed that of native pdb 's distribution, which we believe is not meaningful.
> >
> > Note, we are working on the experiment ProteinWeaver w/ alignment + triangular attention. We expected to obtain the results in next few days.
> >
> > |                 |                                                    |    500   |           |         |    600   |           |         |    800   |           |         |
> > |-----------------|----------------------------------------------------|:--------:|:---------:|:-------:|:--------:|:---------:|:-------:|:--------:|:---------:|:-------:|
> > |                 | model                                              |   scTM   | diversity | novelty |   scTM   | diversity | novelty |   scTM   | diversity | novelty |
> > | Backbone design | Native PDBs                                        | 0.97     | 0.8       |    --   | 0.94     | 0.77      | 0.67    | 0.92     | 0.8       |    --   |
> > |                 | RFdiffusion                                        | 0.79     | 0.89      | 0.67    | 0.66     | 0.99      | 0.67    | 0.49     | 1         | 0.66    |
> > |                 | Proteus                                            | 0.9      | $\color{red}{\text{0.34}}$      | 0.72    | 0.89     | $\color{red}{\text{0.34}}$      | 0.68    | 0.67     | $\color{red}{\text{0.56}}$     | 0.66    |
> > |                 | CarbonNovo w/o plm                                 | 0.41     | 0.76      |    --   | 0.35     | 0.8       |    --   | 0.25     | 1         |    --   |
> > |                 | ProteinWeaver w/o alignment                        | 0.78     | 0.76      | 0.67    | 0.69     | 0.76      | 0.69    | 0.54     | 0.73      | 0.67    |
> > |                 | ProteinWeaver w/o alignment + triangular attention | 0.8      | 0.72      | 0.68    | 0.69     | 0.76      | 0.68    |          |           |         |
> > |                 | ProteinWeaver w/ alignment                         | **0.86** | 0.77      | 0.67    | **0.79** | 0.76      | 0.68    | **0.68** | 0.73      | 0.67    |
> > |                 | ProteinWeaver w/ alignment+ triangular attention   |          |           |         |          |           |         |          |           |         |
> > | Co-design       | Multiflow*                                         | 0.83     | 0.67      | 0.68    | 0.61     | 0.62      | 0.71    | 0.37     |    0.54   |    --   |
> > |                 | CarbonNovo*                                        | 0.85     | 0.67      | 0.68    | 0.87     | 0.93      | 0.7     | 0.52     |     1     |   0.67  |
> >
> > $\ast$ highlight the codesign methods

---

> ### Author Response · Authors · 2024-11-22
> **Rebuttal by Authors**
>
> **[Q11]** For pretraining, does the model start from scratch or pretrained FrameDiff weights? Also, it would be helpful if the authors could provide further details on the training and sampling time: How long does it take to pretrain the model and how many epochs does it run? How long does it take to align the model?
>
> **[A11]** **It should be noted that we did not use Framediff's pretrained weights.** This is because in stage2 we will use the distance map spliced in stage1 as a condition, which is different from the original Framediff. We trained on the 8 V100 32GB NVIDIA GPUs for 10 days and 100 epochs. When aligning the model, we trained for 1 day on 8 V100 32GB NVIDIA GPUs.
>
> **[Q12]** Figure 9: How does ProteinWeaver perform here?
>
> **[A12]** It is worth noting that ProteinWeaver is a protein design model with a controllable number of domains, so it is not suitable to be placed in Figure 9. Figure 9 just wants to illustrate that the number of domains increases as the length of natural proteins increases, but many models do not pay attention to this.
>
> **[Q13]** How does ProteinWeaver perform as the number of domains increases?
>
> **[A13]** We generated more domains at length 600, that is, by assembling three proteins of length 200 into a protein of length 600. **We found that ProteinWeaver supports the generation of three domains with comparable performance as two domain assemblies**, and its generation quality is shown in the table below.
>
> |   length   | multidomain |    scTM   |   scRMSD  |
> |:----------:|:-----------:|:---------:|:---------:|
> | Length 600 |   300+300   | 0.80±0.10 | 4.93±2.31 |
> |            | 200+200+200 | 0.82±0.09 | 4.68±2.15 |

---

### Official Review · Reviewer_zjPL · 2024-11-03

**Soundness:** 3
**Presentation:** 2
**Contribution:** 3
**Rating:** 6
**Confidence:** 3

**Summary:**

The paper introduces ProteinWeaver, a method for protein design. The authors propose a two step “divide-and-assembly” approach, first generating local structures and then assembling them using a diffusion model which is fine-tuned with pairwise comparisons. They evaluate their model empirically and provide comparisons to other state-of-art approaches. They find that ProteinWeaver outperforms other approaches, especially for longer protein chains.

**Strengths:**

- The paper proposes a novel approach (to my knowledge) and evaluates the method empirically with promising results that should be considered for publication.

**Weaknesses:**

- The paper is mostly well written, however, due to a lot of condensed information, it can be hard to follow at some times and leaving a lot of details in the appendix.
- The authors did not perform a statistical analysis of their results. For example, in Table 3 some numbers are rather close. A test for statistical significance could strenghten the results.

**Questions:**

- How does ProteinWeaver generalize in the wild? Did your method find any new interesting protein structures applicable to real problems?

Minor Comments:
- Figure 3 B: Rfdiffusion should have an uppercase F in the right plot.
- Sect. 2.3 could use some proofreading. The sentences in that paragraph are hard to read. E.g. “For each domain, we use f generates corresponding domain.”

---

> ### Author Response · Authors · 2024-11-22
> **Rebuttal by Authors**
>
> We greatly appreciate your constructive feedback regarding several aspects of our manuscript, including concerns about implementation details and statistical analysis of the results. In response, we have implemented two major improvements:
>
> We have supplemented more implementation details in the revised manuscipt.
> We are testing the statistic significance of table 3, and we expect to update the result in the next day.
> We believe these revisions have substantially strengthened the manuscript and welcome your assessment of these changes.
>
>
> **[Q1]** The paper is mostly well written, however, due to a lot of condensed information, it can be hard to follow at some times and leaving a lot of details in the appendix.
>
> **[A1]** We acknowledge your concern about the density of information and accessibility of the manuscript. We have improved the readability by adding more explanatory content to clarify complex concepts.
>
> **[Q2]** The authors did not perform a statistical analysis of their results. For example, in Table 3 some numbers are rather close. A test for statistical significance could strengthen the results.
> We appreciate this suggestion regarding statistical analysis. We agree with the reviewer that this information is crucial for interpreting our results. IWe are testing the statistic significance of table 3, and we expect to update the result in the next day.
>
> **[Q3]** How does ProteinWeaver generalize in the wild? Did your method find any new interesting protein structures applicable to real problems?
>
> **[A3]** Many thanks for the question. To address this concern, we provided a **systematic generalizability analysis for domain assembly using various structures**.
>
>  **we provide an analysis to illustrate the impact of structural variations in domains on assembly quality**. We clustered natural domains in the CATH dataset based on their topological differences, resulting in approximately 530 domain structures that represent a **natural distribution of protein domains**. This dataset allows us to evaluate the effects of structural variations effectively.
>
> We performed **pairwise assemblies of all these domains** and assessed the quality of the designed structures using the scTM score. Our analysis included a **comparison of secondary structure ratios at the assembly interface**, which we believe directly affects domain assembly.
>
> The results presented in Figure 11 in Appendix confirm our expectations: both the type and ratio of secondary structures significantly influence the designability of assembled structures. Stable secondary structures, such as alpha helices and beta strands, are optimal for domain assembly. We found that increasing the ratio of either helices or strands at the interface enhances designability. For instance, as the helix ratio increases from 0% to 90%, the designability measure (scTM) rises from 0.5 to 0.7. Similarly, for beta strands, the scTM increases from 0.56 to 0.63.
>
> **Helices are easier to assemble than strands, with improvements in designability becoming more pronounced as the ratio of helices increases. In contrast, loops—being unstable—significantly impair assembly quality, with scTM dropping from 0.87 to 0.46. This observation provides valuable insights into the principles of native domain assembly.**
>
> **[Q4]** Real-World Applications:
>
> **[A4]** We have successfully applied ProteinWeaver to design functional proteins for practical applications, including substrate-targeting enzymes and bispecific antibodies. These case studies, detailed in Section 3.4, demonstrate our method's capability to address real biological challenges and create novel protein structures with specific functionalities.

---

### Official Review · Reviewer_7CWX · 2024-11-03

**Soundness:** 3
**Presentation:** 2
**Contribution:** 3
**Rating:** 6
**Confidence:** 4

**Summary:**

The paper proposes an innovative framework for designing protein backbones. Inspired by natural protein evolution, the method divides the protein design problem into two stages: domain generation and assembly. ProteinWeaver demonstrates improved performance over existing methods, particularly for longer protein chains, and shows potential for designing proteins with cooperative functions.

**Strengths:**

- The divide-and-assembly strategy is a creative and effective method for protein backbone design, offering a new direction for protein engineering.
- The method shows significant improvements over current state-of-the-art, especially in the design of long-chain proteins, addressing a critical need in the field.

**Weaknesses:**

- The paper could provide more insight into how robust ProteinWeaver is to variations in protein structures, especially those with atypical domain arrangements or those that deviate significantly from known protein families.
- While ProteinWeaver shows strong performance for long-chain proteins, it would be beneficial to see how it performs across a wider range of protein lengths, particularly shorter chains where domain assembly might be less complex but still challenging.

**Questions:**

See weaknesses I've listed.

---

> ### Author Response · Authors · 2024-11-22
> **Rebuttal by Authors**
>
> We greatly appreciate your constructive feedback regarding several aspects of our manuscript, including concerns about structural variance for protein design, the performance of ProteinWeaver for short-chain assembly. In response, we have implemented two major improvements:
> 1. We have provided insightful analysis for the impact of structural variance on domain assembly.
> 2. We have provided experimental results showing the performance for short-chain assembly.
>
> We believe these revisions have substantially strengthened the manuscript and welcome your assessment of these changes.
>
> **[Q1]** The paper could provide more insight into how robust ProteinWeaver is to variations in protein structures, especially those with atypical domain arrangements or those that deviate significantly from known protein families.
>
>
> We thank the reviewer for bringing this concern to our attention. To answer this question, **we provide an analysis of structural variation Impact**.
>
> **we provide an analysis to illustrate the impact of structural variations in domains on assembly quality**. We clustered natural domains in the CATH dataset based on their topological differences, resulting in approximately 530 domain structures that represent a **natural distribution of protein domains**. This dataset allows us to evaluate the effects of structural variations effectively.
>
> We performed **pairwise assemblies of all these domains** and assessed the quality of the designed structures using the scTM score. Our analysis included a **comparison of secondary structure ratios at the assembly interface**, which we believe directly affects domain assembly.
>
> The results presented in Figure 11 in Appendix confirm our expectations: both the type and ratio of secondary structures significantly influence the designability of assembled structures. Stable secondary structures, such as alpha helices and beta strands, are optimal for domain assembly. We found that increasing the ratio of either helices or strands at the interface enhances designability. For instance, as the helix ratio increases from 0% to 90%, the designability measure (scTM) rises from 0.5 to 0.7. Similarly, for beta strands, the scTM increases from 0.56 to 0.63.
>
> **Helices are easier to assemble than strands, with improvements in designability becoming more pronounced as the ratio of helices increases. In contrast, loops—being unstable—significantly impair assembly quality, with scTM dropping from 0.87 to 0.46. This observation provides valuable insights into the principles of native domain assembly.**

---

> ### Author Response · Authors · 2024-11-22
> **Rebuttal by Authors**
>
> **[Q2]** While ProteinWeaver shows strong performance for long-chain proteins, it would be beneficial to see how it performs across a wider range of protein lengths, particularly shorter chains where domain assembly might be less complex but still challenging.
>
> **[A2]** We thank the reviewer for the question. To answer this problem, we provided experimental results using ProteinWeaver to assembly shorter chains. The conclusion is for short-chain protein backbones (100-300 residues), all methods demonstrate the ability to generate high-quality backbones. RFdiffusion shows slightly superior design quality, while ProteinWeaver achieves the highest novelty scores, both with and without alignment.
>
> Table showing the mean results for 100, 200, 300
> |                 | model                                              | scTM | scRMSD | Interface scTM | diversity | novelty |
> |-----------------|----------------------------------------------------|------|--------|----------------|-----------|---------|
> | Backbone design | Native PDBs                                        | 0.97 |  0.72  |      0.86      |    0.77   |    --   |
> |                 | RFdiffusion                                        | **0.98** |  **0.71**  |       0.9      |    0.54   |   0.72  |
> |                 | Proteus                                            | 0.94 |   1.2  |      0.87      |    $\color{red}{\text{0.43}}$   |   0.75  |
> |                 | CarbonNovo w/o plm                                 | 0.59 |  7.07  |      0.47      |    0.69   |   0.69  |
> |                 | ProteinWeaver w/o alignment                        | 0.87 |  2.34  |      0.72      |    0.71   |   **0.64**  |
> |                 | ProteinWeaver w/o alignment + triangular attention | 0.89 |   2.1  |      0.75      |    0.70    |   0.66  |
> |                 | ProteinWeaver w/ alignment                         | 0.92 |  1.53  |      0.75      |    **0.72**   |   **0.64**  |
> | Co-design       | Multiflow*                                         | 0.96 |  1.62  |      0.91      |    0.44   |   0.71  |
> |                 | CarbonNovo*                                        | 0.93 |  1.22  |      **0.95**      |    0.51   |   0.71  |

---

> > ### Comment · Reviewer_7CWX · 2024-11-23
> >
> > Thanks for your rebuttal. I have no further questions and I'll keep my score. I do like this idea.

---

> > > ### Author Response · Authors · 2024-11-23
> > >
> > > Thank you for reading our rebuttal and for your positive feedback. We're glad you like the idea!

---

### Official Review · Reviewer_tR5n · 2024-11-03

**Soundness:** 3
**Presentation:** 2
**Contribution:** 1
**Rating:** 3
**Confidence:** 4

**Summary:**

This work proposes a 2-stage protein backbone generation framework that assembles multiple domains using conditional diffusion model trained on diagonally concatenated distance map of individual domains. The authors perform preference alignment on scTM to further increase the quality of designs.

**Strengths:**

1. First propose 2-stage generation framework for long protein design, which enables the generation of multiple (interacting) domains.
2. Suggest that generation of long proteins fall short due to lack of modeling inter-domain interaction; Reports the number of domains and interface scTM vs. native proteins. It is convincing and interesting argument.

**Weaknesses:**

I believe the paper’s contribution can be divided into two aspects: a 2-stage generation approach and an alignment.

1. **2-stage generation approach:** **The results doesn’t really show that proposed 2-stage approach is effective.** To truly show that 2-stage assembly approach is better than 1-stage approach, I would compare ProteinWeaver vs scTM-aligned RFDiffusion, or ProteinWeaver w/o alignment vs. RFdiffusion. Any model can be additionally aligned to scTM, and it is not surprising that aligned model shows better results.
    1. Appendix Table 3 shows the ProteinWeaver w/o alignment vs. RFdiffusion, and I think it is performing worse than RFDiffusion. sc metrics are only slightly better but diversity is much lower. It just seems like a tradeoff between diversity and designability.
2. **Alignment: Technical novelty and contribution is limited.** I believe preference alignment is already widely used in protein design tasks, and is more like a supplementary analysis within broader papers (as in FoldFlow2 [1]) not a full conference paper, especially when you don’t extensively discuss on an alignment method and draw meaningful discussions from them. The discussion on alignment method is weak, for example,
    1. Ablation study only compares with SFT. But in most cases SFT is precondition for later alignment not the standalone alignment method. How does SFT + PPO performs? [2]
    2. SPPO and many preference alignment methods are tailored to model human preference, i.e. it is hard to decide an absolute, objective ‘preferredness’ of a response, so you bypass that via pairwise comparison. Can you clarify the reasoning behind the preference alignment when we have explicit reward function (i.e. scTM) we want to optimize?
3. **Results**: Even with this additional alignment, performance improvements are marginal. There is big decrease in diversity, and the designability cannot match those of the native PDBs.
4. **Unclear writing**:
    1. line 199-200:

        > To assemble generated domains, we utilize FrameDiff, the SE(3) diffusion model to pretrain the domain assembly model.
        >

        My understanding is that ProteinWeaver uses FrameDiff architecture for domain assembly model. Is this correct? Could you revise this sentence and Appendix (line 690-693) to make it clear that ProteinWeaver uses FrameDiff architecture? Also do you use pretrained weights of FrameDiff?

    2. line245-246 Is it correct to say maximizing p(S) is different from maximizing the quality of generated structures? Isn’t native PDBs have high scTM and low scRMSD?

[1] Huguet, Guillaume, et al. "Sequence-Augmented SE (3)-Flow Matching For Conditional Protein Backbone Generation." arXiv preprint arXiv:2405.20313 (2024).
[2] Ethayarajh, Kawin, et al. Human-centered loss functions (halos). Technical report, Contextual AI, 2023.

**Questions:**

line 245 typo maximine → maximize

---

> ### Author Response · Authors · 2024-11-22
> **Rebuttal by Authors**
>
> We greatly appreciate your constructive feedback regarding several aspects of our manuscript, including concerns about the effectiveness of the two-stage design, ablation study of alignment, designability, and diversity tradeoff. In response, we have implemented four major improvements:
> 1. We have provided experimental evidence demonstrating the effectiveness of the two-stage design.
> 2. We have provided a comprehensive ablation study about the alignment methods and the reward assignment.
> 3. We have provided a detailed and insightful discussion of the tradeoff between designability and diversity.
> We believe these revisions have substantially strengthened the manuscript and welcome your assessment of these changes.
>
> **[Q1]** 2-stage generation approach: The results doesn’t really show that proposed 2-stage approach is effective. To truly show that 2-stage assembly approach is better than 1-stage approach, I would compare ProteinWeaver vs scTM-aligned RFDiffusion, or ProteinWeaver w/o alignment vs. RFdiffusion. Any model can be additionally aligned to scTM, and it is not surprising that aligned model shows better results.
>
> **[A1]** Many thanks for the question: To address the reviewer's concern regarding the effectiveness of the two-stage assembly approach, we conducted a comparison of ProteinWeaver without alignment against RFdiffusion for long-chain structure generation. The two-stage assembly specifically targets long-chain protein design, making this comparison relevant.
>
> The results are summarized in the table below and demonstrate that ProteinWeaver without alignment achieves competitive design quality compared to RFdiffusion:
>
> - At 500 residues: ProteinWeaver scTM score: 0.78 ± 0.14 vs. RFdiffusion: 0.79 ± 0.19
>
> - At 600 residues: ProteinWeaver scTM score: 0.69 ± 0.18 vs. RFdiffusion: 0.66 ± 0.19
>
> - At 800 residues: ProteinWeaver scTM score: 0.54 ± 0.12 vs. RFdiffusion: 0.49 ± 0.12
>
> While RFdiffusion shows slightly higher scores at 500 residues, **ProteinWeaver (w/o alignment) outperforms RFdiffusion at 600 and 800 residues. As the sequence length increases, the performance advantage of ProteinWeaver becomes more pronounced, indicating that the two-stage generation strategy is particularly effective for long-chain backbone design.**
>
> **[Q2]** Appendix Table 3 shows the ProteinWeaver w/o alignment vs. RFdiffusion, and I think it is performing worse than RFDiffusion. sc metrics are only slightly better but diversity is much lower. It just seems like a tradeoff between diversity and designability.
>
> **[A2]** We thank the reviewer for this important observation regarding the relationship between diversity and designability metrics.
> First, regarding the tradeoff between scTM and diversity, this relationship has been well-documented in Table 3 of ProteinBench [1]. there is a consistent inverse relationship between structural quality and diversity across different backbone design methods. **We maintain that structural quality should be prioritized as the primary metric, as diversity and novelty metrics are only meaningful when the generated structures maintain sufficient quality. High diversity scores for structures with poor quality may simply indicate the generation of unrealistic or physically implausible conformations.**
>
> To address the concern about RFdiffusion's higher diversity scores in long-chain backbone design, **we established a benchmark using real protein structures as a golden standard.** We randomly sampled 100 high-resolution experimental structures from the Protein Data Bank (PDB), iteratively removing structures with the highest TM-score compared to others until we obtained a set of 100 distinct structures. **This approach provides a representative snapshot of the actual structural diversity present in natural single-chain proteins.**
>
> **Our analysis, as shown in Figure 3, reveals that even randomly sampled native PDB structures achieve a diversity score of approximately 0.8, which is comparable to ProteinWeaver's diversity but significantly lower than RFdiffusion's scores. This suggests that RFdiffusion's higher diversity scores may not necessarily represent better performance in capturing the distribution training set.**

---

> ### Author Response · Authors · 2024-11-22
> **Rebuttal by Authors**
>
> **[Q3]** Alignment: Technical novelty and contribution is limited. I believe preference alignment is already widely used in protein design tasks, and is more like a supplementary analysis within broader papers (as in FoldFlow2 [1]) not a full conference paper, especially when you don’t extensively discuss on an alignment method and draw meaningful discussions from them. The discussion on alignment method is weak, for example,1. SPPO and many preference alignment methods are tailored to model human preference, i.e. it is hard to decide an absolute, objective ‘preferredness’ of a response, so you bypass that via pairwise comparison. Can you clarify the reasoning behind the preference alignment when we have explicit reward function (i.e. scTM) we want to optimize?
>
> **[A3]** We appreciate the reviewer’s insightful comments. To address this concern, we would like to clarify our rationale regarding the use of scTM score as a precondition in our alignment.
>
> **The computational cost associated with calculating scTM scores is prohibitively high for efficient use in the PPO (Proximal Policy Optimization) framework.** The scTM calculation involves generating eight sequences for each designed backbone using proteinMPNN, followed by structure prediction with ESMFold for each sequence. The sequence with the highest scTM score is then selected for further analysis. As both the number of sampled sequences and the length of the designed proteins increase, the computational demands rise significantly, as illustrated in the table below.
>
> | length |      | seq_per_sample  | (unit: second) |     |
> |--------|:----:|:---------------:|:--------------:|:---:|
> |        | 1    | 2               | 4              | 8   |
> | 50     | 9.9  | 12.3            | 16.3           | 23  |
> | 100    | 10.9 | 12              | 17.2           | 25  |
> | 200    | 16   | 21.3            | 32             | 52  |
> | 300    | 30   | 41              | 66             | 113 |
> | 400    | 51   | 76              | 132            | 216 |
> | 500    | 89   | 130             | 211            | 377 |
>
>  Given this analysis, we concluded that **integrating SFT (Supervised Fine-Tuning) with PPO for structure design is not feasible due to these constraints.** We recognize the importance of alignment methods and appreciate the suggestion for further exploration of SFT + PPO performance, but the computational limitations prevent us from pursuing this combination in our current study.

---

> ### Author Response · Authors · 2024-11-22
> **Rebuttal by Authors**
>
> **[Q4]** Ablation study only compares with SFT. But in most cases SFT is precondition for later alignment not the standalone alignment method. How does SFT + PPO performs?
>
> **[A4]** We thank the reviewer for raising these important points. We agree that comprehensive ablation studies are crucial for assessing the technical contributions of our study, and we have conducted extensive analyses on the preference alignment component.
>
> 1. **Evaluation of Preference Alignment Methods**
> **We evaluated the performance of SPPO with different preference alignment methods: SFT (Supervised Fine-Tuning) and SFT + DPO (Direct Preference Optimization).** To ensure the robustness of our implementation, we tested DPO with various beta parameters. The beta parameter controls the degree of difference between the fine-tuned model and the reference model. A larger beta value makes it less likely for the model to deviate from the reference model during the fine-tuning process.
>
> The results, shown in the updated Table 6 & 7 in the revised manuscript, demonstrate that **SPPO-based alignment consistently outperforms both SFT and SFT + DPO in terms of design quality and novelty across different protein lengths. Notably, the performance advantage of SPPO increases significantly with longer protein sequences, indicating its effectiveness for long-chain protein design.** While SFT + DPO with a beta of 0.1 enhances the diversity of designed structures, it leads to a marked decline in design quality as measured by scTM and scRMSD. This finding highlights an **inverse relationship between structural quality and diversity: as structural quality decreases, diversity tends to increase.**
>
> --Mean result of short length:
> | model              | length100,200,300 |               |           |               |
> |--------------------|:-----------------:|:-------------:|:---------:|:-------------:|
> |                    | scTM              | scRMSD        | diversity | novelty       |
> | SPPO               | **0.91±0.14**     | 1.27±1.84     | 0.59      | **0.61±0.07**     |
> | SFT                | **0.91±0.12**     | **1.20±1.79** | 0.61      | 0.62±0.06     |
> | SFT+DPO (beta=10)  | 0.90±0.11         | 1.45±1.96     | **0.62**  | 0.62±0.04     |
> | SFT+DPO (beta=1)   | 0.88±0.14         | 1.89±2.18     | **0.62**  | 0.62±0.05     |
> | SFT+DPO (beta=0.1) | 0.88±0.14         | 1.89±2.18     | **0.62**  | 0.63±0.05 |
>
> --Mean result of long length:
>
> | model              | length 500, 600, 800 |         |           |          |
> |--------------------|:--------------------:|:-------:|:---------:|:--------:|
> |                    | scTM                 |  scRMSD | diversity | novelty  |
> | SPPO               | **0.72**             | **6.9** | 0.75      | **0.68**     |
> | SFT                | 0.55                 | 16.58   | 0.88      | 0.69     |
> | SFT+DPO (beta=10)  | 0.58                 | 14.92   | 0.87      | 0.7      |
> | SFT+DPO (beta=1)   | 0.66                 | 9.64    | 0.71      | 0.71     |
> | SFT+DPO (beta=0.1) | 0.38                 | 22.24   | **0.99**  | 0.75 |

---

> > ### Author Response · Authors · 2024-11-22
> > **Part2 for Q4**
> >
> > 2. **Different reward assignments**
> >
> > **We explored the impact of varying reward assignments by adjusting preferences from scTM to interface scTM, as well as combining both metrics.** For the combined approach, we selected the sample that exhibited the best performance in both scTM and interface scTM. We believe that scTM serves as a general quality metric for backbone structures, while interface scTM focuses on optimizing inter-domain interactions. The results, presented in the accompanying tables (Table 7 in the revised manuscript), indicate that for various protein lengths, **structures optimized using scTM yield superior quality compared to those optimized with interface scTM.** This suggests that global quality alignment via scTM is more effective than local interface alignment.
> >
> > Additionally, **using a combined reward assignment of scTM and interface scTM resulted in a slight improvement in interface scTM scores compared to using scTM**, with only marginal changes in overall scTM scores. This suggests that **scTM is sufficient for optimizing structural quality.**
> >
> >
> > | model                                         | short length (mean result of 100, 200, 300) |          |                |           |          |
> > |-----------------------------------------------|:-------------------------------------------:|:--------:|:--------------:|:---------:|:--------:|
> > |                                               | scTM                                        |  scRMSD  | interface scTM | diversity | novelty  |
> > | ProteinWeaver (scTM) w/o bo3                  |                   **0.89**                  |   2.07   |      0.74      |    0.71   | **0.64** |
> > | ProteinWeaver (interface scTM) w/o bo3        |                     0.84                    |   2.74   |      0.66      |    0.71   |   0.65   |
> > | ProteinWeaver (scTM + interface scTM) w/o bo3 |                     0.88                    | **1.97** |    **0.76**    |  **0.72** | **0.64** |
> >
> >
> > | model                                         | long length (mean result of 500, 600, 800) |         |                |           |          |
> > |-----------------------------------------------|:------------------------------------------:|:-------:|:--------------:|:---------:|:--------:|
> > |                                               | scTM                                       |  scRMSD | interface scTM | diversity | novelty  |
> > | ProteinWeaver (scTM) w/o bo3                  |                    0.72                    | **6.9** |    **0.66**    |    0.75   | **0.68** |
> > | ProteinWeaver (interface scTM) w/o bo3        |                    0.67                    |   9.46  |       0.6      |  **0.82** |   0.69   |
> > | ProteinWeaver (scTM + interface scTM) w/o bo3 |                  **0.73**                  |   6.94  |    **0.66**    |    0.76   | **0.68** |

---

> ### Author Response · Authors · 2024-11-22
> **Rebuttal by Authors**
>
> **[Q5]** Results: Even with this additional alignment, performance improvements are marginal and the designability cannot match those of the native PDBs.
>
> **[A5]** Thanks for the question. **There is a misunderstanding for the reviewer. Acturally, none of any existing backbone design methods can match the designability of native PDBs. To comprehensively demonstrate ProteinWeaver's advanced performance for long-chain protein design. We have compared ProteinWeaver against more state-of-the-art methods, including Proteus, Multiflow, and CarbonNovo w/o protein language model.** It is important to note that Multiflow and CarbonNovo are co-design methods that recruit additional sequence information for training. Therefore, the comparison is not fair. However, we still recruit these baselines to demonstrate ProteinWeaver's superior performance for long chain backbone design.
>
> 1. For short-chain protein backbones (100-300 residues), all methods demonstrate the ability to generate high-quality backbones, consistent with our original manuscript. RFdiffusion shows slightly superior design quality, while ProteinWeaver achieves the highest novelty scores, both with and without alignment.
>
> Table showing the mean results for 100, 200, 300
> |                 | model                                              | scTM | scRMSD | Interface scTM | diversity | novelty |
> |-----------------|----------------------------------------------------|------|--------|----------------|-----------|---------|
> | Backbone design | Native PDBs                                        | 0.97 |  0.72  |      0.86      |    0.77   |    --   |
> |                 | RFdiffusion                                        | **0.98** |  **0.71**  |       0.9      |    0.54   |   0.72  |
> |                 | Proteus                                            | 0.94 |   1.2  |      0.87      |    $\color{red}{\text{0.43}}$   |   0.75  |
> |                 | CarbonNovo w/o plm                                 | 0.59 |  7.07  |      0.47      |    0.69   |   0.69  |
> |                 | ProteinWeaver w/o alignment                        | 0.87 |  2.34  |      0.72      |    0.71   |   **0.64**  |
> |                 | ProteinWeaver w/o alignment + triangular attention | 0.89 |   2.1  |      0.75      |    0.70    |   0.66  |
> |                 | ProteinWeaver w/ alignment                         | 0.92 |  1.53  |      0.75      |    **0.72**   |   **0.64**  |
> | Co-design       | Multiflow*                                         | 0.96 |  1.62  |      0.91      |    0.44   |   0.71  |
> |                 | CarbonNovo*                                        | 0.93 |  1.22  |      **0.95**      |    0.51   |   0.71  |

---

> ### Author Response · Authors · 2024-11-22
> **Part 2 for Q5**
>
> 2. In the case of long-chain protein backbones (500-800 residues), **ProteinWeaver demonstrates significantly higher scTM scores compared to RFdiffusion and CarbonNovo (without protein language model). Furthermore, ProteinWeaver outperforms co-design methods Multiflow for long-chain backbones and CarbonNovo for 800-length backbones. This unfair comparison demonstrate ProteinWeaver 's advantage for long-chain backbone design.**
>
> 3. **While Proteus achieves the best design quality metrics (scTM and scRMSD), it shows significantly reduced structural diversity compared to other methods.** Case analysis reveals that Proteus generates structures limited to three categories, primarily helical tandem repeats, confirming our diversity metric findings. **Based on these observations, we conclude that ProteinWeaver offers the most balanced performance for long-chain backbone design, successfully combining high designability with high novelty.**
>
>
> |                 |                                                    |    500   |           |         |    600   |           |         |    800   |           |         |
> |-----------------|----------------------------------------------------|:--------:|:---------:|:-------:|:--------:|:---------:|:-------:|:--------:|:---------:|:-------:|
> |                 | model                                              |   scTM   | diversity | novelty |   scTM   | diversity | novelty |   scTM   | diversity | novelty |
> | Backbone design | Native PDBs                                        | 0.97     | 0.8       |    --   | 0.94     | 0.77      | 0.67    | 0.92     | 0.8       |    --   |
> |                 | RFdiffusion                                        | 0.79     | 0.89      | 0.67    | 0.66     | 0.99      | 0.67    | 0.49     | 1         | 0.66    |
> |                 | Proteus                                            | 0.9      | $\color{red}{\text{0.34}}$      | 0.72    | 0.89     | $\color{red}{\text{0.34}}$      | 0.68    | 0.67     | $\color{red}{\text{0.56}}$     | 0.66    |
> |                 | CarbonNovo w/o plm                                 | 0.41     | 0.76      |    --   | 0.35     | 0.8       |    --   | 0.25     | 1         |    --   |
> |                 | ProteinWeaver w/o alignment                        | 0.78     | 0.76      | 0.67    | 0.69     | 0.76      | 0.69    | 0.54     | 0.73      | 0.67    |
> |                 | ProteinWeaver w/o alignment + triangular attention | 0.8      | 0.72      | 0.68    | 0.69     | 0.76      | 0.68    |          |           |         |
> |                 | ProteinWeaver w/ alignment                         | **0.86** | 0.77      | 0.67    | **0.79** | 0.76      | 0.68    | **0.68** | 0.73      | 0.67    |
> |                 | ProteinWeaver w/ alignment+ triangular attention   |          |           |         |          |           |         |          |           |         |
> | Co-design       | Multiflow*                                         | 0.83     | 0.67      | 0.68    | 0.61     | 0.62      | 0.71    | 0.37     |    0.54   |    --   |
> |                 | CarbonNovo*                                        | 0.85     | 0.67      | 0.68    | 0.87     | 0.93      | 0.7     | 0.52     |     1     |   0.67  |
>
> $\ast$ highlight the codesign methods
>
> Based on these results, we have demonstrated ProteinWeaver's superior performance in designing long-chain protein structures compared to other backbone design methods. A key innovation of ProteinWeaver is its incorporation of the hierarchical, multi-domain nature of protein backbones into its modeling approach, representing an important contribution to the field of protein design.

---

> ### Author Response · Authors · 2024-11-22
> **Rebuttal by Authors**
>
> **[Q6]** Unclear writing:
>   1. line 199-200:To assemble generated domains, we utilize FrameDiff, the SE(3) diffusion model to pretrain the domain assembly model. My understanding is that ProteinWeaver uses FrameDiff architecture for domain assembly model. Is this correct? Could you revise this sentence and Appendix (line 690-693) to make it clear that ProteinWeaver uses FrameDiff architecture? Also do you use pretrained weights of FrameDiff?
>
> **[A6]** Thank you for seeking clarification about our relationship with FrameDiff. We apologize for any confusion in our manuscript. To clarify: while our domain assembly model shares architectural similarities with FrameDiff's SE(3) diffusion framework, **we do not use FrameDiff's pretrained weights**. We will revise both the main text (lines 199-200) and the Appendix (lines 690-693) to clearly state that our model adopts a similar SE(3) diffusion architecture but is trained independently with our own weights.
>
> **[Q7]** line245-246 Is it correct to say maximizing p(S) is different from maximizing the quality of generated structures? Isn’t native PDBs have high scTM and low scRMSD?
>
> **[A7]** Native PDBs typically exhibit favorable quality metrics (high scTM, low scRMSD), and it is reasonable to maximize p(S) during model training. During the alignment, we used generated structure to fine-tune the model, therefore, we used quality matrics as a guidance for the winner and loser structure.
>
> **[Q8]** line 245 typo maximine → maximize
>
> **[A8]** We thank the reviewer for pointing the typo, and we have corrected this type in the revised manuscript.

---

### Official Review · Reviewer_eGhm · 2024-11-05

**Soundness:** 2
**Presentation:** 2
**Contribution:** 2
**Rating:** 3
**Confidence:** 4

**Summary:**

The paper introduces ProteinWeaver, a two-stage framework for protein backbone design. In the first stage, individual protein domains are generated, and in the second stage, these domains are assembled using an SE(3) diffusion model. Additionally, the framework employs preference alignment to discern complex relationships between structural and interaction landscapes, aiming to enhance performance.

**Strengths:**

The proposed method outperforms Chroma, RFDiffusion, FrameDiff, and FrameFlow in generating long backbone structures.

**Weaknesses:**

1. The comparative evaluation is insufficient. While the authors emphasize that their method achieves superior performance on long proteins compared to Chroma and RFDiffusion, other recent methods—such as MultiFlow [1], CarbonNovo [2], Proteus [3], and FoldFlow2 [4]—have also demonstrated better performance on long proteins. To substantiate their claims and highlight the main contributions, the authors should include these methods in their comparisons.
2. The Related Work section does not adequately reflect the current literature on protein backbone generation. At a minimum, the methods mentioned in Weakness 1 should be discussed to provide a comprehensive overview of recent advancements.
3. I don't think the basic ideas are well-motivated. The proposed method assumes that any domains generated in the first stage can be assembled, which may not hold true concerning both function and structure.
4. The methodology lacks sufficient novelty. It consists of two main components: a diffusion model for assembling domains and preference alignment to discern structural and interaction landscapes. Both components appear to be straightforward applications of well-established methods in the literature, raising concerns about the originality of the approach.

[1] Campbell et al. Generative Flows on Discrete State-Spaces: Enabling Multimodal Flows with Applications to Protein Co-Design. Proceedings of the 41st International Conference on Machine Learning. https://proceedings.mlr.press/v235/campbell24a.html

[2] Ren et al. CarbonNovo: Joint Design of Protein Structure and Sequence Using a Unified Energy-based Model. Proceedings of the 41st International Conference on Machine Learning. https://proceedings.mlr.press/v235/ren24e.html

[3] Wang et al. Proteus: Exploring Protein Structure Generation for Enhanced Designability and Efficiency. Proceedings of the 41st International Conference on Machine Learning. https://proceedings.mlr.press/v235/wang24bi.html

[4] Huguet et al.  Sequence-Augmented SE(3)-Flow Matching For Conditional Protein Backbone Generation. https://arxiv.org/abs/2405.20313

**Questions:**

1. Please address the points in Weakness section.

2.  In the domain assembly generation stage, the authors indicate that a linker is inserted between adjacent domains, which may increase the overall protein length. How does the method ensure that the generated protein aligns with the target length?

3. What is the split configuration used during testing? Does the performance rely on this configuration? For example, assembling a 200-residue domain with a 300-residue domain for a target protein of 500 residues might yield different results compared to combining a 100-residue domain with a 400-residue domain.

4. Section 2.2 mentions that multi-domain proteins are filtered out, despite the method’s aim to model domain-domain interactions. What is the rationale behind excluding multi-domain proteins when they are essential for studying inter-domain interactions?

---

> ### Author Response · Authors · 2024-11-22
> **Rebuttal by Authors**
>
> We greatly appreciate your constructive feedback regarding several aspects of our manuscript, including concerns about insufficient baseline comparisons, the overview of recent advancements, our methodological motivation, and the depth of ablation studies. In response, we have implemented four major improvements:
> 1. We have expanded our experimental evaluation to include additional baseline methods for a more comprehensive performance comparison.
> 2. We have thoroughly revised the related work section to better reflect the current state of literature on protein backbone generation.
> 3. We have provided a more detailed analysis to clearly articulate our motivation and contributions.
> 4. We have provided more ablation studies to study the performance of different alignment methods and reward assignments.
> We believe these revisions have substantially strengthened the manuscript and welcome your assessment of these changes.

---

> ### Author Response · Authors · 2024-11-22
> **Rebuttal by Authors**
>
> **[Q1]** The comparative evaluation is insufficient. While the authors emphasize that their method achieves superior performance on long proteins compared to Chroma and RFDiffusion, other recent methods—such as MultiFlow, CarbonNovo, Proteus, and FoldFlow2—have also demonstrated better performance on long proteins. To substantiate their claims and highlight the main contributions, the authors should include these methods in their comparisons.
>
> **[A1]** We appreciate the reviewer’s concern regarding the comparative evaluation. In response, we have **expanded our analysis** to include additional state-of-the-art methods in both the backbone design (Proteus) and sequence-structure codesign (CarbonNovo and MultiFlow) categories. **It is important to note that sequence-structure codesign methods leverage both sequence and structure datasets for model training, which makes direct comparisons with our backbone design methods—focused solely on backbone information—unfair. Nonetheless, we have included comparisons with these methods.** Additionally, FoldFlow2 was not included in our analysis as its source code is not publicly available.
>
> We provide a brief introduction of these methods. A key limitation shared by all of these approaches is that they fail to account for the hierarchical, multi-domain nature of long-chain proteins.
>
> 1. **Proteus:** A state-of-the-art backbone design method that employs AlphaFold2's graph-based triangle approach and multi-track interaction networks. It outperforms RFdiffusion in long-chain monomer (>400 residues) generation and exceeds Chroma in complex structure generation.
>
> 2. **MultiFlow:** A sequence-structure codesign method using discrete flow-based generative models. It demonstrates superior designability compared to RFdiffusion, partly due to its **knowledge distillation from ProteinMPNN.**
>
> 3. **CarbonNovo:** An end-to-end sequence-structure codesign method using a unified energy-based model. It pioneered the **integration of protein language models (pLM)** for both structure and sequence generation, showing substantial improvement in designability (from 51.75% to 81.38%).
>
> More detailed comparative results of this analysis are presented in Table 3 & 4 of the revised manuscript.
>
> 1. For short-chain protein backbones (100-300 residues), all methods demonstrate the ability to generate high-quality backbones, consistent with our original manuscript. RFdiffusion shows slightly superior design quality, while ProteinWeaver achieves the highest novelty scores, both with and without alignment.
>
> Table showing the mean results for 100, 200, 300
> |                 | model                                              | scTM | scRMSD | Interface scTM | diversity | novelty |
> |-----------------|----------------------------------------------------|------|--------|----------------|-----------|---------|
> | Backbone design | Native PDBs                                        | 0.97 |  0.72  |      0.86      |    0.77   |    --   |
> |                 | RFdiffusion                                        | **0.98** |  **0.71**  |       0.9      |    0.54   |   0.72  |
> |                 | Proteus                                            | 0.94 |   1.2  |      0.87      |    $\color{red}{\text{0.43}}$   |   0.75  |
> |                 | CarbonNovo w/o plm                                 | 0.59 |  7.07  |      0.47      |    0.69   |   0.69  |
> |                 | ProteinWeaver w/o alignment                        | 0.87 |  2.34  |      0.72      |    0.71   |   **0.64**  |
> |                 | ProteinWeaver w/o alignment + triangular attention | 0.89 |   2.1  |      0.75      |    0.70    |   0.66  |
> |                 | ProteinWeaver w/ alignment                         | 0.92 |  1.53  |      0.75      |    **0.72**   |   **0.64**  |
> | Co-design       | Multiflow*                                         | 0.96 |  1.62  |      0.91      |    0.44   |   0.71  |
> |                 | CarbonNovo*                                        | 0.93 |  1.22  |      **0.95**      |    0.51   |   0.71  |

---

> ### Author Response · Authors · 2024-11-22
> **Extended response for Q1**
>
> 2. **In the case of long-chain protein backbones (500-800 residues), ProteinWeaver demonstrates significantly higher scTM scores compared to RFdiffusion, MultiFlow, and CarbonNovo (without protein language model). Additionally, ProteinWeaver outperforms CarbonNovo (with protein language model) in scTM for 800 residue backbone design.**
>
> 3. While **Proteus achieves the best design quality metrics (scTM and scRMSD), it shows significantly reduced structural diversity compared to other methods** (highlighted in $\color{red}{\text{red}}$). Case analysis reveals that Proteus generates structures limited to three categories, primarily helical tandem repeats, confirming our diversity metric findings.
>
> 4. An inverse relationship between scTM with diversity is observed. For 600 and 800 length, RFdiffusion's scTM decreases and diversity increases. The diversity score even exceed that of native pdb 's distribution, which we believe is not meaningful.
>
> Note, we are working on the experiment ProteinWeaver w/ alignment + triangular attention. We expected to obtain the results in next few days.
>
> |                 |                                                    |    500   |           |         |    600   |           |         |    800   |           |         |
> |-----------------|----------------------------------------------------|:--------:|:---------:|:-------:|:--------:|:---------:|:-------:|:--------:|:---------:|:-------:|
> |                 | model                                              |   scTM   | diversity | novelty |   scTM   | diversity | novelty |   scTM   | diversity | novelty |
> | Backbone design | Native PDBs                                        | 0.97     | 0.8       |    --   | 0.94     | 0.77      | 0.67    | 0.92     | 0.8       |    --   |
> |                 | RFdiffusion                                        | 0.79     | 0.89      | 0.67    | 0.66     | 0.99      | 0.67    | 0.49     | 1         | 0.66    |
> |                 | Proteus                                            | 0.9      | $\color{red}{\text{0.34}}$      | 0.72    | 0.89     | $\color{red}{\text{0.34}}$      | 0.68    | 0.67     | $\color{red}{\text{0.56}}$     | 0.66    |
> |                 | CarbonNovo w/o plm                                 | 0.41     | 0.76      |    --   | 0.35     | 0.8       |    --   | 0.25     | 1         |    --   |
> |                 | ProteinWeaver w/o alignment                        | 0.78     | 0.76      | 0.67    | 0.69     | 0.76      | 0.69    | 0.54     | 0.73      | 0.67    |
> |                 | ProteinWeaver w/o alignment + triangular attention | 0.8      | 0.72      | 0.68    | 0.69     | 0.76      | 0.68    |          |           |         |
> |                 | ProteinWeaver w/ alignment                         | **0.86** | 0.77      | 0.67    | **0.79** | 0.76      | 0.68    | **0.68** | 0.73      | 0.67    |
> |                 | ProteinWeaver w/ alignment+ triangular attention   |          |           |         |          |           |         |          |           |         |
> | Co-design       | Multiflow*                                         | 0.83     | 0.67      | 0.68    | 0.61     | 0.62      | 0.71    | 0.37     |    0.54   |    --   |
> |                 | CarbonNovo*                                        | 0.85     | 0.67      | 0.68    | 0.87     | 0.93      | 0.7     | 0.52     |     1     |   0.67  |
>
> $\ast$ highlight the codesign methods

---

> ### Author Response · Authors · 2024-11-22
> **Rebuttal by Authors**
>
> **[Q2]** The Related Work section does not adequately reflect the current literature on protein backbone generation. At a minimum, the methods mentioned in Weakness 1 should be discussed to provide a comprehensive overview of recent advancements.
>
> **[A2]** We thank the reviewer for bringing this concern to our attention. In response, we have substantially expanded and updated the Related Work section to provide a comprehensive overview of recent advancements in protein backbone generation. We also recruited sequence-structure co-design methods Multiflow and CarbonNovo. The revised section now includes detailed discussions of recent methodological developments and better reflects the current state of the field.

---

> ### Author Response · Authors · 2024-11-22
> **Rebuttal by Authors**
>
> **[Q3 and Q4]** I don't think the basic ideas are well-motivated. The proposed method assumes that any domains generated in the first stage can be assembled, which may not hold true concerning both function and structure.
>
> The methodology lacks sufficient novelty. It consists of two main components: a diffusion model for assembling domains and preference alignment to discern structural and interaction landscapes. Both components appear to be straightforward applications of well-established methods in the literature, raising concerns about the originality of the approach.
>
> **[A3 and A4]** We thank the reviewer for raising these questions. We have provided a combined response to address them, and have also modified the revised manuscript to better articulate the motivation and contributions of our study.
>
> 1. **Clarification of Contributions**
>  There appears to be a **misunderstanding** regarding the contributions of our study. **Domain assembly is a well-established long-history task in protein design that has been extensively researched using traditional computational biology methods, such as Rosetta.**
>
>  To clarify, we provide the following examples that highlight the significance of this task:
>
>   - **Domain-Assembly for Structural Design**: A recent study demonstrated the use of helical protein blocks for designing extendable nanomaterials [1]. This work serves as a proof of concept for the domain-based design approach, but it focused solely on pure helical-bundle assemblies. In contrast, our study generalizes this idea to encompass a broader range of domains with structural variations. We present the first application of deep learning methods for general domain assembly, marking an important new paradigm in this field.
>
>   - **Domain-Assembly for Functional Design**: Nearly two decades ago, research showed that well-assembled domains, through interface-directed evolution, could dramatically enhance both affinity and specificity—achieving over 500-fold and 2,000-fold increases, respectively [2]. More recently, another study revealed that assembling a substrate recruitment domain to an enzyme significantly improved its functional specificity [3].
>
> These studies underscore the validity and importance of our motivation for pursuing domain-assembly-based protein structure and function design, a concept we introduced in our manuscript. We have added more references in the revised manuscript to emphasize this importance.
>
> [1] Huddy T F, Hsia Y, Kibler R D, et al. Blueprinting extendable nanomaterials with standardized protein blocks[J]. Nature, 2024, 627(8005): 898-904.
>
> [2] Huang J, Koide A, Makabe K, et al. Design of protein function leaps by directed domain interface evolution[J]. Proceedings of the National Academy of Sciences, 2008, 105(18): 6578-6583.
>
> [3] Park R, Ongpipattanakul C, Nair S K, et al. Designer installation of a substrate recruitment domain to tailor enzyme specificity[J]. Nature chemical biology, 2023, 19(4): 460-467.
>
> 2. **Clarification on Domain Assembly**
>
> There appears to be a **misunderstanding**. **We did not claim that any domain can be assembled. Instead, we asserted that domains can be assembled to generate proteins designed for both structural and functional purposes.**
>
> 3. **Analysis of Structural Variation Impact**
>
> To further clarify our approach, **we provide an analysis to illustrate the impact of structural variations in domains on assembly quality**. We clustered natural domains in the CATH dataset based on their topological differences, resulting in approximately 530 domain structures that represent a **natural distribution of protein domains**. This dataset allows us to evaluate the effects of structural variations effectively.
>
> We performed **pairwise assemblies of all these domains** and assessed the quality of the designed structures using the scTM score. Our analysis included a **comparison of secondary structure ratios at the assembly interface**, which we believe directly affects domain assembly.
>
> The results presented in Figure 11 in Appendix confirm our expectations: both the type and ratio of secondary structures significantly influence the designability of assembled structures. Stable secondary structures, such as alpha helices and beta strands, are optimal for domain assembly. We found that increasing the ratio of either helices or strands at the interface enhances designability. For instance, as the helix ratio increases from 0% to 90%, the designability measure (scTM) rises from 0.5 to 0.7. Similarly, for beta strands, the scTM increases from 0.56 to 0.63.
>
> **Helices are easier to assemble than strands, with improvements in designability becoming more pronounced as the ratio of helices increases. In contrast, loops—being unstable—significantly impair assembly quality, with scTM dropping from 0.87 to 0.46. This observation provides valuable insights into the principles of native domain assembly.**

---

> > ### Author Response · Authors · 2024-11-22
> > **Part 2 for Q3 and Q4**
> >
> > In response to the reviewer's concerns, we have also experimented with adding triangular attention to ProteinWeaver and evaluated its performance. We replaced the edge transition from MLP to triangular attention, and the performance results are shown in the updated table in Section Appendix Figure 8. **Our findings indicate that ProteinWeaver's two-stage is a general framework that can incorporate triangular attention, which significantly enhances the design quality of ProteinWeaver.**
> >
> > Please note that we are still testing the results for long-chain proteins and anticipate providing the complete results in the coming days.
> >
> > | Model                               | length100     |               | length200     |               | length300     |               |
> > |-------------------------------------|---------------|---------------|---------------|---------------|---------------|---------------|
> > |                                     | scTM          | scRMSD        | scTM          | scRMSD        | scTM          | scRMSD        |
> > | ProteinWeaver(MLP)                  | 0.90±0.10     | 1.73±1.83     | 0.85±0.10     | 2.36±1.52     | 0.86±0.09     | 3.07±2.21     |
> > | ProteinWeaver(Triangular attention) | **0.92±0.11** | **1.66±1.70** | **0.88±0.10** | **1.99±1.47** | **0.88±0.09** | **2.64±1.96** |

---

> > > ### Comment · Reviewer_eGhm · 2024-11-23
> > >
> > > I appreciate your efforts in the rebuttal and have read through all your responses. However, upon reviewing the revised manuscript, I noticed that you have disclosed your names and affiliations, which violates the double-blind review policy. If the policy permits, I will continue to provide further comments on your response.

---

> > > > ### Author Response · Authors · 2024-11-23
> > > > **Response**
> > > >
> > > > We apologize for our carelessness. We have edited the pdf to correct the mistake.

---

> ### Author Response · Authors · 2024-11-22
> **Rebuttal by Authors**
>
> **[Q5]** In the domain assembly generation stage, the authors indicate that a linker is inserted between adjacent domains, which may increase the overall protein length. How does the method ensure that the generated protein aligns with the target length?
>
> **[A5]** Thank you for this important question about length control in our domain assembly process. To clarify: when we insert linkers between adjacent domains, these **linker lengths are included in our total length calculations**. The target protein length specified in the input represents the total length including both domains and linkers, ensuring that our final generated structure precisely matches the requested length specification.
>
> **[Q6]** What is the split configuration used during testing? Does the performance rely on this configuration? For example, assembling a 200-residue domain with a 300-residue domain for a target protein of 500 residues might yield different results compared to combining a 100-residue domain with a 400-residue domain.
>
> **[A6]** Thank you for your question. To address it, we investigated **the impact of split configurations on design performance** by testing two lengths: 400 and 600 residues, using various configurations. The results, presented in the table below and updated in revised manuscript, indicate that **different configurations do have a slight effect on performance**.
>
> | **length**      |             | **scTM**  | **scRMSD** |
> |-----------------|-------------|-----------|------------|
> | **Length 400** | **200+200** | 0.89±0.06 | 2.48±1.15  |
> |                 | **100+300** | 0.88±0.06 | 2.86±1.55  |
> | **length 600** | **100+500** | 0.77±0.12 | 5.89±3.11  |
> |                 | **200+400** | 0.79±0.13 | 5.14±3.96  |
> |                 | **300+300** | 0.80±0.10 | 4.93±2.31  |
>
> **[Q7]** Section 2.2 mentions that multi-domain proteins are filtered out, despite the method’s aim to model domain-domain interactions. What is the rationale behind excluding multi-domain proteins when they are essential for studying inter-domain interactions?
>
> **[A7]** Thank you for highlighting this potential source of confusion. We would like to clarify that there appears to be a misunderstanding in our manuscript's wording. **We do not filter out multi-domain proteins; rather, we specifically filter and select multi-domain proteins for training to study domain-domain interactions.** We will revise it the manuscript clearly indicate that multi-domain proteins are integral to our training process, as they are essential for studying inter-domain interactions.

---

### Author Response · Authors · 2024-11-22
**We are pleased to hear any feedback from all reviewers before discussion deadline.**

Dear Reviewers, ACs, and SACs,
We would like to sincerely thank all the reviewers for their efforts and valuable suggestions. We have made every effort to address the reviewers' concerns, including the following:
1. Provided sufficient evaluation on more baseline methods.
2. We have expanded the related work section in our revised manuscript.
3. Included rational explanations highlighting both the scientific and methodology contributions of our study.
4. Corrected minor presentation issues.
We appreciate everyone's time and effort in providing insightful feedback, which has greatly helped us improve our manuscript. We have revised the paper to incorporate many of the reviewers' suggestions and comments, and we are truly grateful for your contributions.
We welcome any further feedback during the discussion phase!
Thank you once again!
Best regards, Authors

---

### Note · Program_Chairs · 2024-11-23
**Submission Desk Rejected by Program Chairs**

The revision uploaded at 22 Nov 2024 at 14:27 Eastern Standard Time contained the author names on the first page of the PDF, which was seen by the reviewers. The paper is desk rejected for violating the anonymity policy and breaking double blind review. The decision was confirmed by the program chairs.